# Organosulfate Produced from Consumption of SO₃ Speeds up Sulfuric Acid-Dimethylamine Atmospheric Nucleation

Xiaomeng Zhang[1], Yongjian Lian[1], Shendong Tan[1], Shi Yin*[1]

[1]MOE & Guangdong Province Key Laboratory of Laser Life Science & Institute of Laser Life Science, Guangzhou Key Laboratory of Spectral Analysis and Functional Probes, College of Biophotonics, South China Normal University, Guangzhou 510631, P. R. China

*Correspondence to*: Shi Yin (yinshi@m.scnu.edu.cn)

**Abstract.** Although sulfuric acid (SA) and dimethylamine (DMA) driven nucleation mainly dominants the new particle formation (NPF) process in the atmosphere, seeking the involvement of other gaseous species remains crucial to better understand the NPF. Organosulfate has been detected in gas phase and abundantly in atmospheric fine particles. However, its molecular formation mechanism and its impact on the NPF are still much less understood. Here, we explored the gas phase reaction of Glycolic acid (GA) with $SO_3$, and evaluated the enhancing potential of its products on the SA-DMA driven NPF using a combination of quantum chemical calculations and kinetics modeling. We found that the considerable concentration of glycolic acid sulfate (GAS) is thermodynamically accessible from the reaction of GA with $SO_3$, efficiently catalyzed by SA or $H_2O$ molecules. The produced GAS can form stable clusters with SA and DMA, and speeds up the nucleation rate of SA-DMA system obviously. Notably, the enhancement by GAS on the SA-DMA-based particle formation rate can be up to ~ 800 times in the region where the concentration of SA is about $10^4$ molecules $cm^{-3}$. Supported by observations of atmospheric NPF events at Mt. Tai in China, our proposed ternary GAS-SA-DMA nucleation mechanism further indicates that the organosulfates produced from the consumption of $SO_3$ may play an important role for the unexpected high NPF rates observed in areas with relatively low concentrations of SA. The presented reaction and nucleation mechanisms provide a new feasible source of organosulfates in atmospheric new particles. Based on our findings, the impact of organosulfates on the atmospheric NPF in multiple regions around the world was estimated and discussed.

## 1 Introduction

Atmospheric aerosols have a significant influence on global climate, local air quality, and human health (Wang et al., 2015;Wang et al., 2020;Lee et al., 2019;Zhang et al., 2004;Rose et al., 2018). New particle formation (NPF) in the atmosphere, including formation of critical nucleus and subsequent growth of the nuclei, accounts for a significant fraction of secondary organic aerosols (SOA) (Rose et al., 2018;Yao et al., 2018;Zhang, 2010;Feketeová et al., 2019;Kirkby et al., 2016). It has been widely accepted that sulfuric acid (SA) is one of the most important nucleation precursors in the atmosphere (Zhang et al., 2004;Almeida et al., 2013;Kirkby et al., 2011;Loukonen et al., 2010;Zhao et al., 2011;Sipila et al., 2010;Zhang et al., 2022b;Olenius et al., 2017;Lehtipalo et al., 2016). Dimethylamine (DMA) was found to be one of the strongest species to

enhance SA-driven NPF (Olenius et al., 2017;Yao et al., 2018). However, the binary SA-DMA-driven NPF still could not fully explain the observed NPF events globally, since there is still a gap between the observed particle formation rates and simulated rates (Kirkby et al., 2016;Shen et al., 2019;Liu et al., 2021b;Shen et al., 2020). Therefore, seeking the involvement of other gaseous species to better understand the NPF has been paid extensive attentions in recent years (Olenius et al., 2017;Shen et al., 2019;Tan et al., 2022a;Ehn et al., 2014;Kawamura and Bikkina, 2016;Zhang et al., 2004).

Organic species commonly detected in aerosols are thought to play significant roles in particle formation and direct and indirect aerosol forcing (Jimenez et al., 2009;O'Dowd et al., 2002). Organosulfates have been identified as the most abundant class of organosulfur compounds, accounting for 5−30% of the organic mass fraction in atmospheric particles (Brüggemann et al., 2017;Tolocka and Turpin, 2012;Shakya and Peltier, 2015;Froyd et al., 2010;Mutzel et al., 2015;Glasius et al., 2018). Katz et al. measured the presence of organosulfates and identified its importance to new particle formation (Katz et al., 2023). Ehn et al. have made the first observation of gas phase glycolic acid sulfate (GAS) in the Finnish forest (Ehn et al., 2010), and GAS has been characterized and identified as the most abundant organosulfates in the fine particles collected from southeastern USA (Hettiyadura et al., 2017). Various organosulfates have also been identified and characterized from fine particulate matter samples collected in the United States (Hettiyadura et al., 2015), China (Wang et al., 2018), Mexico City, and Pakistan (Olson et al., 2011). These observed organosulfates represented by GAS and lactic acid sulfate are suggested to originate from the reaction of a variety of volatile organic compounds (Froyd et al., 2010;Darer et al., 2011;Riva et al., 2015;Kundu et al., 2013;Zhang et al., 2012a;Zhang et al., 2014). However, despite extensive research and significant progress (Riva et al., 2015;Riva et al., 2016;Passananti et al., 2016;Ye et al., 2018;Zhu et al., 2019;McNeill, 2015;McNeill et al., 2012), huge areas of uncertainty remain in the current understanding of organosulfates, their molecular formation pathways and their impacts on the atmospheric NPF (Brüggemann et al., 2020).

Organic acids, which are frequently observed in the atmosphere, have been expected to participate in the process of atmospheric nucleation, with a focus on the thermochemical properties of clusters between organic acids and common atmospheric nucleation precursors (Zhang et al., 2022). Glycolic acid (GA) as the simplest α-hydroxy acid is a highly oxidized multifunctional organic acid and has been detected in diverse environments with relatively high concentration (Miyazaki et al., 2014;Mochizuki et al., 2019;Brüggemann et al., 2017). For example, Mochizuki et al. have found that the maxima of GA in gas phase could be up to 343 ng m$^{-3}$ at Mt. Tai in the north China (Mochizuki et al., 2017), and shown that the gas-phase concentrations of total monoacids including GA and lactic acid are higher than those of particle-phase. Miyazaki et al. have detected the presence of GA in the gas phase within the marine atmospheric boundary layer over the western subarctic North Pacific (Miyazaki et al., 2014). Interestingly, organic compounds with -OH or -COOH group have been proven to have highly reactive activity with $SO_3$ (Li et al., 2018;Mitsui et al., 2011;Zhuang and Pavlish, 2012), which is a major air pollutant and mainly emitted from the gas-phase oxidation of $SO_2$ (Mitsui et al., 2011;Chen and Bhattacharya, 2013;Cao et al., 2010;Zhong et al., 2018). Due to the presence of polar functional groups (-OH and -COOH groups), the two active sites of α-hydroxy acid

can react with $SO_3$ to form carboxylic acid sulfates and carboxylic acid sulfuric anhydrides, individually (Tan et al., 2022b;Liu et al., 2019;Mackenzie et al., 2015;Smith et al., 2019;Smith et al., 2017, 2018;Shen et al., 1990). Recent computational study has probed the clusters formation mechanism of GA-SA and $NH_3$ and identified that GA acts as a mediate bridge for the formation of SA-$NH_3$-based clusters (Zhang et al., 2017). Organic sulfur species, mainly including organosulfates (Nguyen et al., 2014) and carboxylic sulfuric anhydrides (Zhang et al., 2018b;Rong et al., 2020;Zhang et al., 2022a), with relatively lower vapor pressure have also been inferred to facilitate the occurrence of NPF in the atmosphere. However, although many studies have illustrated that atmospheric organic species produced by gas-phase chemical reaction can exert significant influence over atmospheric NPF processes (Zhang et al., 2012b;Liu et al., 2018a;Wang et al., 2010;Hirvonen et al., 2018;Laaksonen et al., 2008;Ristovski et al., 2010;Metzger et al., 2010), the cluster formation mechanism of GAS, produced from the reaction of GA and $SO_3$, with the typical nucleation precursors SA and DMA have never been systematically investigated and compared.

In current research, we performed a comparative study on GA as well as its products (GAS and GASA) to probe the role of organic acid, organosulfate, and organic sulfuric anhydride in enhancing the SA-DMA nucleation potential by evaluating the formation mechanism of GA-SA-DMA, GAS-SA-DMA, and GASA-SA-DMA systems. We have obtained the minimum free energy structures of the $(GA)_x(SA)_y(DMA)_z$, $(GAS)_x(SA)_y(DMA)_z$ and $(GASA)_x(SA)_y(DMA)_z$ $(0 \leq z \leq x + y \leq 3)$ systems. Kinetics of the clusters formation pathways and rates were obtained via the Atmospheric Cluster Dynamics Code (ACDC) simulations, which use the calculated thermodynamic data of acquired clusters as input (McGrath et al., 2012;Olenius et al., 2013). The simulated particle formation rates of GAS involved system were compared with those of GA and GASA involved systems based on correspondingly observational concentration (Stieger et al., 2021;Miyazaki et al., 2014;Mochizuki et al., 2019;Mochizuki et al., 2017). Additionally, we also compared the calculated ternary GAS-SA-DMA nucleation rates with the field observations of NPF at Mt. Tai in China, and found that the ternary nucleation mechanism involved GAS was well supported by the observations. Finally, the impact of organosulfates on the atmospheric NPF in multiple regions around the world was estimated and discussed.

**2 Method**

**2.1 Configurational sampling**

A multistep global minimum sampling scheme, which has previously been applied to study the atmospheric cluster formation (Ma et al., 2019), was employed to search for the global minima of the $(GA)_x(SA)_y(DMA)_z$, $(GAS)_x(SA)_y(DMA)_z$, and $(GASA)_x(SA)_y(DMA)_z$ $(0 \leq z \leq x + y \leq 3)$ clusters. And the global minimum structures of GA, GAS, and GASA molecules were taken from previous study (Tan et al., 2022b). To locate the global minimum energy structure, the artificial bee colony algorithm was systematically employed by ABCluster program to generate 1000 initial random configurations for each cluster (Zhang and Dolg, 2015), and then these configurations were furtherly pre-optimized using the PM7 semi-empirical method

implemented in the MOPAC2016 program (Stewart, 2013, 2007;MOPAC, 2016). Second, up to 100 structures with relatively lower energies were selected from the 1000 structures, and M06-2X/6-31+G* level of theory was applied for subsequent optimization. Finally, the further geometry optimization and frequency calculations at the M06-2X/6-311++G(3df, 3pd) level of theory were performed to optimize the ten best of 100 optimized configurations, and then the global minimum structure with the lowest energy was obtained.

## 2.2 Quantum chemical calculations

All the density functional theory (DFT) calculations were implemented in the GAUSSIAN 09 program package.(Frisch, 2009) The M06-2X functional combined with 6-311++G(3df, 3pd) basis set was chosen as it has been proven to be accurate to estimate the thermodynamic properties of atmospheric clusters, such as organic acid-SA-amine clusters, amide-SA clusters, amino acid-SA clusters and so on.(Clark et al., 1983;Elm et al., 2012;Herb et al., 2011;Elm et al., 2015;Elm et al., 2016;Ge et al., 2018a;Ge et al., 2018b) Intrinsic reaction coordinate (IRC) calculations were carried out to verify the connections of the transition states with the reactants and products. Single-point energies were calculated at the DLPNO-CCSD(T)/aug-cc-pVTZ level based on the optimized geometries using the ORCA 4.1.2 package (Riplinger and Neese, 2013;Riplinger et al., 2013;Neese, 2012;Lu, 2022), which has gained popularity in the large cluster formation studies (Xie et al., 2017;Chen et al., 2020;Shen et al., 2019). Zero-point energies at the M06-2X/6-311++G(3df,3pd) level were performed to correct for the corresponding single-point energies. The corresponding formation Gibbs free energy of the stable clusters are summarized in the Supporting Information (SI).

## 2.3 The concentration of glycolic acid sulfate (GAS) and glycolic acid sulfuric anhydride (GASA)

The formation of GAS and GASA can be described by following two reactions, respectively.

$$(GA + SO_3)_{-OH} \rightarrow GAS$$

$$(GA + SO_3)_{-COOH} \rightarrow GASA$$

The equilibrium constant $K_{(GA + SO3)-OH}$ for the formation of GAS and $K_{(GA + SO3)-COOH}$ for the formation of GASA are

$$K_{(GA + SO3)-OH} = \frac{[GAS]}{[GA][SO_3]} = e^{\frac{-\Delta G}{RT}}$$

$$K_{(GA + SO3)-COOH} = \frac{[GASA]}{[GA][SO_3]} = e^{\frac{-\Delta G}{RT}}$$

and the equilibrium concentration of GAS and GASA can be roughly estimated theoretically using the following expressions:

$$[GAS] = K_{(GA + SO3)-OH} [GA][SO_3]$$

$$[GASA] = K_{(GA + SO3)-COOH} [GA][SO_3]$$

where $K_{(GA + SO3)-OH}$ and $K_{(GA + SO3)-COOH}$ are equal to the equilibrium constants from the formation Gibbs energies of the GAS

and GASA, respectively. [GA] and [$SO_3$] are the concentration of GA and $SO_3$ monomer, respectively. We use the reactant

concentrations of [GA] = $1.11 \times 10^7$-$2.72 \times 10^9$ molecules cm$^{-3}$ according to the values of some field observations (Mochizuki

et al., 2019; Miyazaki et al., 2014; Stieger et al., 2021; Mochizuki et al., 2017). Considering atmospheric $SO_3$ field

measurements (Yao et al., 2020), its concentration is considered in the range of $10^4$ - $10^6$ molecules cm$^{-3}$. Based on the above

equations, the estimated concentration of the reaction product, GAS, is about $2.14 \times 10^2$-$5.24 \times 10^6$ molecules cm$^{-3}$, and GASA

is about $2.30 \times 10^{-7}$-$5.62 \times 10^{-3}$ molecules cm$^{-3}$. Thus, a range of concentration for GAS, from $10^2$ to $10^6$ molecules cm$^{-3}$, is

selected for the discussion in this work (Figure 3, Table S1, and Figure S9).

**2.4 Atmospheric cluster dynamics code (ACDC) kinetic model**

The Atmospheric Cluster Dynamics Code (ACDC) simulations is a dynamical model where the time development of molecular

cluster concentrations is solved by integrating numerically the birth-death equations using the MATLAB-R2019b program

(McGrath et al., 2012;Olenius et al., 2013;Ortega et al., 2012;Shampine and Reichelt, 1997). In the current research, the ACDC

was employed to investigate the formation pathways and formation rates of the clusters. The birth-death equations can be

written as

$$\frac{dc_i}{dt} = \frac{1}{2}\sum_{j<i} \beta_{j,(i-j)} c_j \, c_{(i-j)} + \sum_j \gamma_{(i+j)\to i} c_{i+j} - \sum_j \beta_{i,j} c_i c_j - \frac{1}{2}\sum_{j<i} \gamma_{i\to j} c_i + Q_i - S_i \qquad (1)$$

where $i$ and $j$ are the clusters given in the system, $c_i$ and $c_j$ are the concentration of cluster $i$ and $j$, $\beta_{i,j}$ is the collision coefficient

between clusters $i$ and $j$, and $\gamma_{(i+j)\to i}$ is the evaporation coefficient of cluster ($i + j$) evaporating into clusters $i$ and $j$. $Q_i$ is the

possible additional sources of cluster $i$, $S_i$ is the sink terms for taking into account external losses of cluster $i$. The collision

coefficient $\beta$ is calculated using the kinetic gas theory as equation 2 (taking $\beta_{i,j}$ as example)

$$\beta_{ij} = \left(\frac{3}{4\pi}\right)^{1/6} \left(\frac{6k_bT}{m_i} + \frac{6k_bT}{m_j}\right)^{1/2} \left(V_i^{1/3} + V_j^{1/3}\right)^2 \qquad (2)$$

where $k_b$ is the Boltzmann constant, $T$ is the temperature, $m_i$ and $m_j$ are the masses of $i$ and $j$, respectively, and $V_i$ and $V_j$ are

their respective volumes. The evaporation coefficient $\gamma_{(i+j)\to i}$ is calculated using the Gibbs free energies of formation of the

clusters

$$\gamma_{(i+j)\to i} = \beta_{ij} \frac{c_i^e c_j^e}{c_{i+j}^e} = \beta_{ij} c_{ref} \, exp\left\{\frac{\Delta G_{i+j} - \Delta G_i - \Delta G_j}{k_bT}\right\} \qquad (3)$$

where $c_i^e$ is the equilibrium concentration of cluster $i$, $\Delta G_i$ is the Gibbs free energy of the formation of cluster $i$, and $c_{ref}$ is the

monomer concentration at the reference vapor corresponding to the pressure of 1 atm at which the Gibbs free energies were

determined.

Here, the ACDC simulation system is regarded as a "3 × 3" box, containing $(GA)_x(SA)_y(DMA)_z$, $(GAS)_x(SA)_y(DMA)_z$,

$(GASA)_x(SA)_y(DMA)_z$ (where the total number of $x$ and $y$ from 0 to 3, $z$ from 0 to 3) clusters. Among these clusters, only the

clusters including an equal number of $z$ and $x + y$ or the clusters with smaller numbers of $z$ and $x + y$ were considered, as only

150 these clusters have the potential to further grow into larger sizes (Li et al., 2018;Olenius et al., 2013).

Detailed description about the boundary clusters that are allowed to leave the simulation and contribute to NPF is presented in the SI. In the current study, the GA concentration [GA] ranged from $10^7$ to $10^{10}$ molecules cm$^{-3}$ and the corresponding GAS concentration [GAS] was $10^3 \sim 10^6$ molecules cm$^{-3}$ (Miyazaki et al., 2014;Mochizuki et al., 2019;Mochizuki et al., 2017;Stieger et al., 2021). A constant coagulation sink of $2.6 \times 10^{-3}$ s$^{-1}$ was applied to account for scavenging by larger particles. The

155 simulations were mainly run at 278 K, with additional runs at 258 K and 298 K to investigate the influence of temperature. These conditions correspond to a typical sink value and temperature in the boreal forest environment (Olenius et al., 2013; Maso et al., 2008).

## 3 Results and discussion

### 3.1 The Reaction of GA with SO$_3$

The potential energy surfaces (PES) of the reaction between GA and SO$_3$, along with the optimized structures of pre-reaction complexes (R), transition states (TS) and products (P) are presented in Figure 1. Since GA contains both –OH and –COOH functional groups, two reaction pathways for GA and SO$_3$ are considered in this study: 1) the esterification reaction between hydroxyl group of GA and SO$_3$ [(GA + SO$_3$)$_{-OH}$]; 2) the cycloaddition reaction between carboxyl group of GA and SO$_3$ [(GA + SO$_3$)$_{-COOH}$]. In the presumed reaction pathway 1) (without catalyst), the hydroxyl oxygen atom of GA could react with the

sulfur atom of SO$_3$ to form sulfate GAS (Figure 1a), followed by simultaneous proton transfer from GA to SO$_3$. But the Gibbs free energy barrier is 23.08 kcal mol$^{-1}$, which shows that a direct reaction between hydroxyl group and SO$_3$ is thermodynamically unfavorable pathway for the formation of GAS. The high energy barrier of (GA + SO$_3$)$_{-OH}$ reaction partly ascribes to the ring tension of rather closed four-membered ring transition state structure. Differently, the Gibbs free energy barrier of (GA + SO$_3$)$_{-COOH}$ reaction path without catalyst is significantly lower, only 3 kcal mol$^{-1}$ (Figure 1b).

With high abundance ($\sim 10^{17}$ cm$^{-3}$) being detected in the troposphere (Huang et al., 2015), H$_2$O has been reported to effectively act as a catalyst in chemical reactions (Liu et al., 2019). Herein, we investigated the catalytic effect of H$_2$O on the reaction of GA and SO$_3$. As can be seen in Figure 1a, the free energy barrier of the (GA + SO$_3$)$_{-OH}$ reaction catalyzed by H$_2$O is 3.41 kcal mol$^{-1}$, which is substantially lower than that for the direct reaction without catalyst. The free energy barrier for the H$_2$O-catalyzed (GA + SO$_3$)$_{-COOH}$ reaction is 0.92 kcal mol$^{-1}$. These results indicate that both reaction pathways for GA + SO$_3$ are

favorable with the catalysis of H$_2$O to generate GAS and GASA, respectively. Therefore, as the relative humidity (RH) increases, it should be conducive to the formation of GAS and GASA. The abundances of hydrated GA clusters GA-$n$(H$_2$O) ($n$ = 0 - 3) were also calculated at different RH (Figure S3 and Table S3). The relative equilibrium abundance of GA hydrates is less than 7% at RH = 90% and 298 K. Since the hydration of GA is weak, the effect of the hydrated GA clusters to the

formation of GAS and GASA is not further considered. SA can also act as both an acceptor and a donor of hydrogen, thereby

promoting various proton transfer reactions to facilitate the formation of GAS/GASA with low barrier (Yao et al., 2018;Liu et al., 2017;Tan et al., 2018). As shown in Figure 1a (black line), the energy barrier of GA + $SO_3$ reaction in the presence of catalyst SA sharply decrease, with the value of 2.83 kcal mol$^{-1}$ for the formation of GAS, and with that of 1.13 kcal mol$^{-1}$ for the formation of GASA (Black line in Figure 1b). As the geometry structures of transition states (TS1 to TS6) shown in Figure 1, the participation of catalysts could efficiently decrease the ring tension of the transition state geometry through increasing

the ring size, followed with the reduction of related energy barrier.

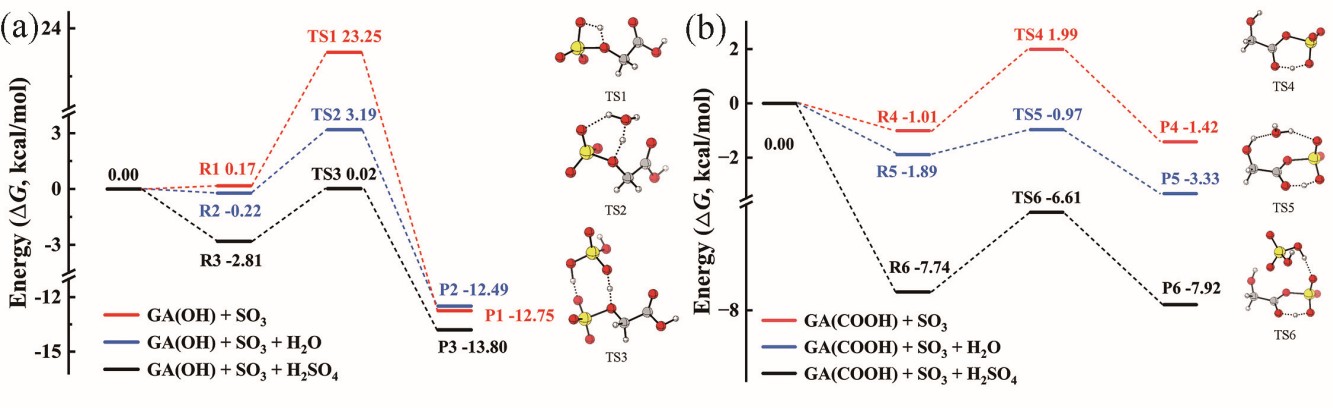

**Figure 1.** Potential energy surfaces at the DLPNO-CCSD(T)/aug-cc-pVTZ//M06-2X/6-311++G(3df,3pd) level of theory in units of kcal mol$^{-1}$ (at 298 K, 1atm) for the gas-phase reactions of GA and $SO_3$ through paths (a) $SO_3$ attacking the -OH group of GA and (b) $SO_3$ attacking the -COOH group of GA. The red line represents the reaction without catalyst; the blue line represents the reaction with $H_2O$ as a catalyst;

and black line represents the reaction with $H_2SO_4$ as a catalyst. R, TS, and P refer to pre-reaction complex, transition state, and product, respectively. Hydrogen, carbon, oxygen, and sulfur atoms are represented by white, gray, red, and yellow spheres, respectively.

Hence, the catalysts $H_2O$ or SA could make the GA + $SO_3$ reaction barrierless and readily to occur in the atmosphere. Indeed, many $SO_3$-involved gas-phase reactions can be effectively catalyzed by $H_2O$ or SA (Li et al., 2018;Liu et al., 2019). Note that previous computational studies have proved that organic acids can act as catalysts to the reaction of $SO_3$ and $H_2O$ to form

sulfuric acid (Hazra and Sinha, 2011). Although we pay more attention to the catalytic effect of $H_2O$ on the reaction of α-hydroxy acid with $SO_3$ in this study, the possible pathway of GA catalyzing $SO_3$ + $H_2O$ → SA reaction should not be ignored. Its PES (black line) is considered and compared with $H_2O$ catalytic reaction paths (see details in Figure S1 and related discussions in the SI). It is worth noting that the product GAS generated from GA and $SO_3$, with a more negative Gibbs free energy of formation (-12.75 kcal mol$^{-1}$), is more stable than the product GASA (-1.42 kcal mol$^{-1}$) as displayed in Figure 1.

This may suggest a possible formation pathway for the gas-phase GAS observed in the atmosphere. To the best of our knowledge, the gas-phase GAS has been detected for the first time in the Finnish boreal forest (Ehn et al., 2010). Le Breton et al. identified and measured 17 sulfur-containing organics (including organosulfates and GAS is one of them) at a regional site 40 km north-west of Beijing (Le Breton et al., 2018). They successfully identified a persistent gas-phase presence of organosulfates in the ambient air. The mean contribution from gas-phase sulfur-containing organics to total was found up to

be 11.6 %, ~23 ng m$^{-3}$. Ye et al. also detected the ion $C_2H_3SO_6^-$ with a diurnal peak in the afternoon in both gas phase and

particle phase, which ion was attributed to GAS, at Guangzhou in southern China during the autumn of 2018 (Ye et al., 2021). Unfortunately, atmospheric field observation data on gas-phase organosulfates and organic sulfuric anhydrides are still relatively scarce. In order to explore their impacts on the atmospheric NPF, we calculated atmospheric concentrations of GAS and GASA based on the thermodynamic equilibrium of the chemical reactions. The estimated concentration of GAS is in the range of $10^3$ - $10^5$ molecules cm$^{-3}$ and that of GASA is ~ $10^{-6}$ - $10^{-4}$ molecules cm$^{-3}$ (see details in Table S1 and the first part of SI).

### 3.2 Cluster thermodynamic data

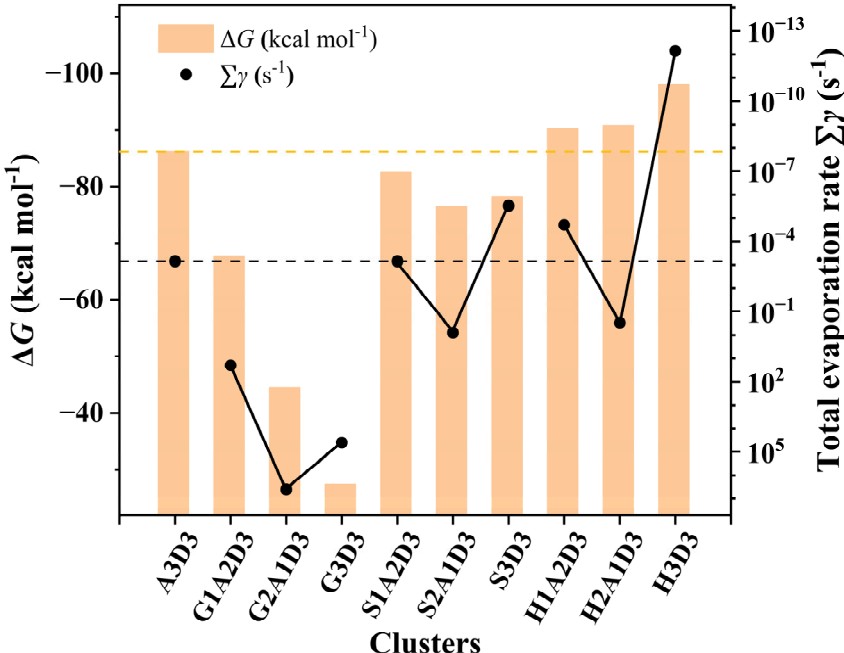

**Figure 2**. The formation Gibbs free energies $\Delta G$ (kcal mol$^{-1}$) and evaporation rates $\sum\gamma$ (s$^{-1}$) of the $(GA)_x(SA)_y(DMA)_3$, $(GAS)_x(SA)_y(DMA)_3$, and $(GASA)_x(SA)_y(DMA)_3$ ($x = 0$-3, $x + y = 3$) clusters calculated at the DLPNO-CCSD(T)/aug-cc-pVTZ//M06-2X/6-311++G(3df,3pd) level of theory and 278 K. DMA, SA, GA, GAS, and GASA are represented by D, A, G, S, and H, individually.

In order to evaluate the enhancing potential of GA and its different reaction products GAS/GASA on the typical SA-DMA driven NPF process, we analysed the global minimal of GA-SA-DMA, GAS-SA-DMA, and GASA-SA-DMA cluster. The identified lowest free energy structures of $(GA)_x(SA)_y(DMA)_z$, $(GAS)_x(SA)_y(DMA)_z$ and $(GASA)_x(SA)_y(DMA)_z$ ($0 \leq z \leq x + y \leq 3$) clusters are depicted in Figure S4, Figure S5 and Figure S6, respectively. In general, both intermolecular hydrogen bond and ion electrostatic interactions formed by proton transfer reactions are found to play key role in stabilizing these clusters. Hydrogen bond interactions are observed in all clusters. The -OH/-COOH group in GA/GAS/GASA participates in at least one hydrogen bond formation and acts as the donor/acceptor of hydrogen. The proton transfer reactions generally occurred in the acid-base clusters are found in most of the DMA-containing heteromolecular clusters, but not observed in the acidic homomolecular $(GA)_x$, $(GAS)_x$, $(GASA)_x$ ($x = 1$-3) clusters and GA-SA, GAS-SA, GASA-SA clusters, in which only hydrogen bonds interaction are found. Interestingly, there is no proton transfer in the $(GA)_1(DMA)_1$, $(GAS)_1(DMA)_1$ and $(GASA)_1(DMA)_1$ clusters. However, when another extra molecule adds to these three clusters, not only the hydrogen bonds

interaction is enhanced, but also the proton transfer is promoted. This is because the trimer or large clusters are sufficient to convert hydrogen-bonded system to ion electrostatic interaction system, a consequence of proton transfer reaction.

The formation Gibbs free energies $\Delta G$ and evaporation rates values of GA-SA-DMA, GAS-SA-DMA, and GASA-SA-DMA clusters are obtained along with the acquisition of global minima structures. More details about $\Delta G$ and evaporation rates are shown in Table S4 - S6, and Figure S7. Previous researches have verified that DMA is one of the strongest compounds for stabilizing SA clusters (Yao et al., 2018;Almeida et al., 2013;Jen et al., 2014). Since the $(SA)_3(DMA)_3$ cluster is the most stable cluster of SA-DMA system, its $\Delta G$ and evaporation rate were taken as a reference value for comparison. In Figure 2, the formation free energies $\Delta G$ (orange histograms) and evaporation rates $\sum\gamma$ (black points) at 278 K of the $(SA)_3(DMA)_3$ cluster are presented along with the $(GA)_x(SA)_y(DMA)_3$, $(GAS)_x(SA)_y(DMA)_3$, and $(GASA)_x(SA)_y(DMA)_3$ ($x = 0\text{-}3$, $x + y = 3$) clusters as a comparison. The $\Delta G$ of $(GA)_{1\text{-}3}(SA)_{0\text{-}2}(DMA)_3$ clusters are in all cases much more positive than that of $(SA)_3(DMA)_3$ cluster, within a difference in the range of 18.49-58.74 kcal mol$^{-1}$. For $(GAS)_{1\text{-}3}(SA)_{0\text{-}2}(DMA)_3$ clusters, their $\Delta G$ are close to that of $(SA)_3(DMA)_3$ cluster, which is slightly negative, and the difference is in the range of 3.61-9.69 kcal mol$^{-1}$. Notably, the $\Delta G$ of $(GASA)_{1\text{-}3}(SA)_{0\text{-}2}(DMA)_3$ clusters become more negative than that of $(SA)_3(DMA)_3$ cluster. Their value discrepancies are -4.13, -4.63 and -11.83 kcal mol$^{-1}$, respectively, with the number of GASA increase from 1 to 3. These above results indicate that the various organic compounds generated via the chemical reaction, such as GA and its products GAS and GASA, can apparently lead to different formation Gibbs free energies of clusters with the same scale, when participating in the nucleation of the SA-DMA system, due to their unequal intermolecular hydrogen bond and ion electrostatic interaction capacities with acidic and basic molecules, respectively.

The comparison of evaporation rates of these clusters is more interesting. For $(Org)_x(SA)_y(DMA)_3$ (Org = GA, GAS and GASA; $x = 1\text{-}3$, $x + y = 3$) clusters, the evaporation rate of $(Org)_x(SA)_y(DMA)_3$ does not simply change with the increasing number of organic molecules. The evaporation rate of $(Org)_2(SA)(DMA)_3$ (Org = GA, GAS and GASA) is the largest, respectively. The evaporation rates of $(GAS)_1(SA)_2(DMA)_3$, $(GAS)_3(DMA)_3$, $(GASA)_1(SA)_2(DMA)_3$ and $(GASA)_3(DMA)_3$ clusters vary from $10^{-13}$ to $10^{-4}$ s$^{-1}$, which are smaller than that of $(SA)_3(DMA)_3$ cluster, implying the substitutions of one or three SA by GAS and GASA molecules are beneficial for stabilizing clusters. However, the evaporation rates of GA-SA-DMA clusters are found to be the largest comparing with that of corresponding GAS-SA-DMA, GASA-SA-DMA and $(SA)_3(DMA)_3$ clusters displayed in Figure 2. Actually, the evaporation rates of all GA-SA-DMA clusters we calculated are larger than $10^1$ s$^{-1}$ (Figure S7), indicating the instability of the participation of GA to the SA-DMA cluster system. The more negative $\Delta G$ and smaller evaporation rates of GAS/GASA-SA-DMA clusters suggest that they are thermodynamically more favourable than GA-SA-DMA system in the cluster formation process, resulting from the greater binding ability between GAS/GASA and SA-DMA system than that between GA and SA-DMA system. Hence, we can make an initial conclusion that GAS/GASA produced from GA + SO$_3$ reaction may efficiently stabilize the SA-DMA system, in contrast to GA itself.

## 3.3 Cluster formation rates

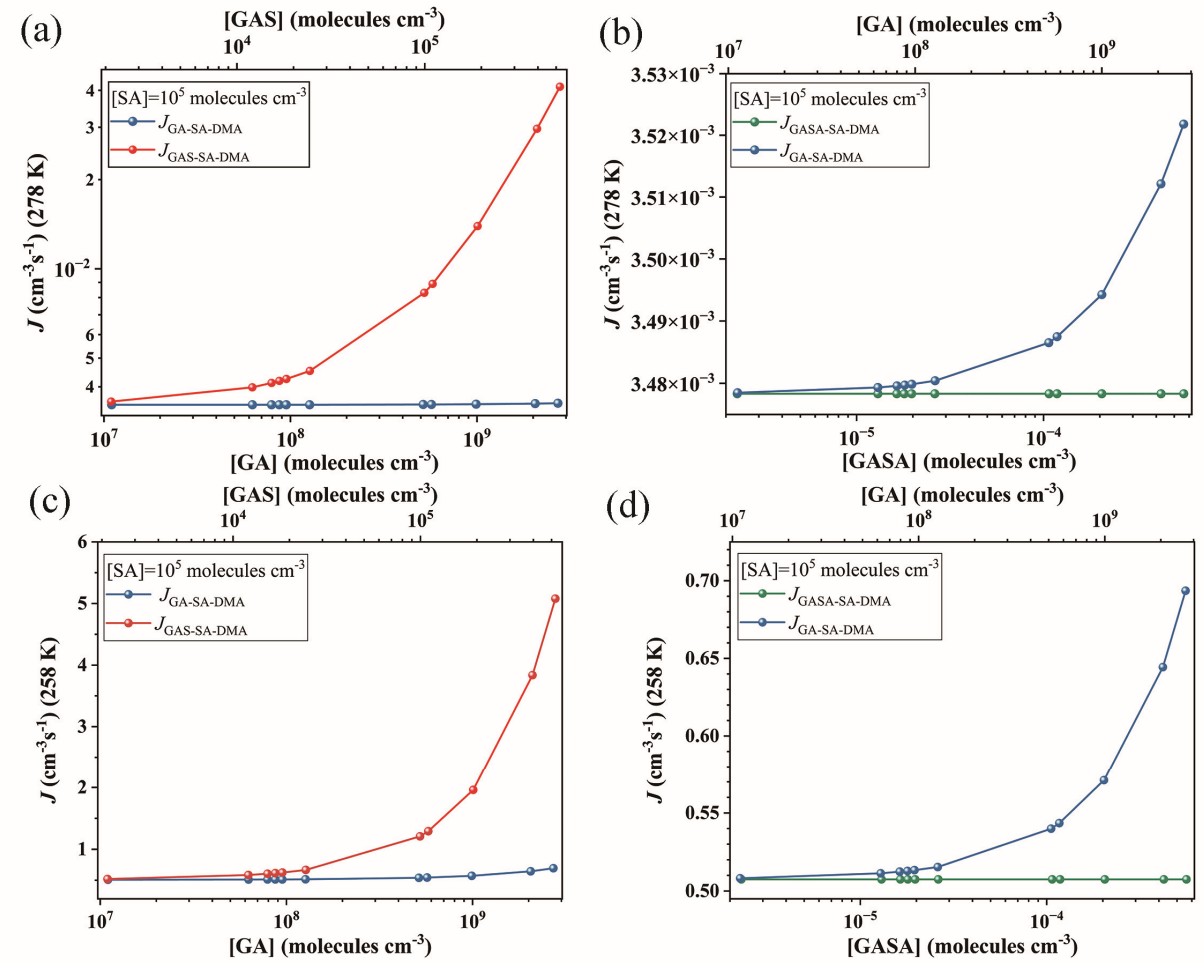

**Figure 3.** Simulated cluster formation rates $J$ (cm$^{-3}$s$^{-1}$) as a function of monomer concentrations ([GA], [GAS], and [GASA], respectively) at (a) (b) 278 K and (c) (d) 258 K under the condition of [DMA] = $10^8$ molecules cm$^{-3}$ and [SA] = $10^5$ molecules cm$^{-3}$. Note that the simulated $J_{\text{GA-SA-DMA}}$ are the same data, but the Y-axis scale are different at (a) (b) and (c) (d), individually.

The corresponding formation rates of the clusters in GA-SA-DMA, GAS-SA-DMA and GASA-SA-DMA systems were further investigated and compared, to achieve a deeper understanding of the influence of GA/GAS/GASA on SA-DMA-based system using ACDC simulations. Figure 3 presents the comparison of the cluster formation rate as a function of the concentration of GA ($1.1 \times 10^7$ - $2.7 \times 10^9$ molecules cm$^{-3}$), GAS ($2.1 \times 10^3$ - $5.2 \times 10^5$ molecules cm$^{-3}$) and GASA ($2.3 \times 10^{-6}$ - $5.6 \times 10^{-4}$ molecules cm$^{-3}$) for the Org-SA-DMA systems at 278 K and 258 K, under the condition of [SA] = $10^5$ molecules cm$^{-3}$. The concentration of DMA is selected to be $10^8$ molecules cm$^{-3}$, according to the typical concentrations observed in the gas phase at high mountains (Liu et al., 2018b, Matsumoto et al., 2023). To compare the enhancing potential of GAS and GA on SA-DMA-based NPF, Figure 3a presents the cluster formation rates as a function of [GAS] and [GA] at 278 K. The cluster formation rate of GAS-SA-DMA system markedly increases with the increasing concentration of [GAS] compared to that of GA-SA-DMA system, especially in the case of [GAS] > $3 \times 10^4$ molecules cm$^{-3}$. As the concentration of GAS increased from $2.1 \times 10^3$ to $5.2 \times 10^5$ molecules cm$^{-3}$, the cluster formation rate of the GAS-SA-DMA system yields a 10-fold increase (Figure 3a), whereas the cluster formation rate of the GA-SA-DMA system basically remains unchanged (from ~ $3.48 \times 10^{-3}$ cm$^{-3}$s$^{-1}$

to ~ $3.52 \times 10^{-3}$ cm$^{-3}$s$^{-1}$, as displayed more clearly in Figure 3b) with the increase of the corresponding concentration of GA. Although the concentration of GAS is typically 4 orders of magnitude lower than that of GA, it has a significantly higher enhancing potential than GA on the SA-DMA-based nucleation system. The cluster formation rate as a function of [GASA] for GASA-SA-DMA system is also compared with that of [GA] for GA-SA-DMA system at 278 K in Figure 3b. The growing trend of cluster formation rate for GASA-SA-DMA system is even smaller than that for GA-SA-DMA system. Combined with the cluster thermodynamic results discussed in the previous section, we can conclude that such different trends should be responsible by the combined impact of cluster $\Delta G$, cluster evaporation rate and organic species concentration of these systems, respectively. The most rapid increase trend of cluster formation rate of GAS-SA-DMA system is a consequence of the favourable $\Delta G$ and small evaporation rates of $(GAS)_x(SA)_y(DMA)_z$ clusters and non-low equilibrium concentration of GAS. The unfavourable $\Delta G$ and large evaporation rates of $(GA)_x(SA)_y(DMA)_z$ clusters make GA-SA-DMA system kinetically unfavourable, even with high GA concentration. For the trend of GASA-SA-DMA system, the extremely low equilibrium concentration of GASA makes its cluster formation rate hard to promote even with the thermodynamically favourable conditions for $(GASA)_x(SA)_y(DMA)_z$ clusters formation. [SO$_3$] = $10^4$ and $10^6$ molecules cm$^{-3}$ are also considered and compared with the results shown in Figure 3a (as displayed in Figure S9). In the case of [SO$_3$] = $10^4$ molecules cm$^{-3}$, it is worth noting that the cluster formation rate of GAS-SA-DMA system slightly increases with the increasing [GAS] compared to that of GA-SA-DMA system with corresponding [GA], which $J_{\text{GAS-SA-DMA}}$ reaches twice the value of $J_{\text{GA-SA-DMA}}$. For [SO$_3$] = $10^6$ molecules cm$^{-3}$, the trend of this difference becomes relatively obvious, and $J_{\text{GAS-SA-DMA}}$ grows up to 2 orders of magnitude higher than $J_{\text{GA-SA-DMA}}$.

The effect of temperature and SA concentration on the cluster formation rates of these systems were also explored. As the temperature decrease to 258 K (Figure 3c and Figure 3d), the cluster formation rates of all considered systems increase by about 2 to 3 orders of magnitude compared to that of 278 K. This behaviour mainly results from that the clusters become thermodynamically favourable under relatively low temperature, making the clusters more stabilized. Simulated cluster formation rates $J$ (cm$^{-3}$s$^{-1}$) of these systems under different [SA] ($10^4$, $10^6$ and $10^7$ molecules cm$^{-3}$) at 278 K are presented in Figure S8. Their increase trends of cluster formation rates as a function of organic species concentration are consistent with that obtained above at [SA] = $10^5$ molecules cm$^{-3}$. The cluster formation rates of these systems also become large with [SA] increase. For example, the highest cluster formation rate of GAS-SA-DMA system is ~ $4 \times 10^{-2}$ cm$^{-3}$s$^{-1}$ within the considered GAS concentration at 278 K when [SA] = $10^5$ molecules cm$^{-3}$ (Figure 3a), while this value could be up to $2.1 \times 10^3$ cm$^{-3}$s$^{-1}$ when [SA] increase to $10^7$ molecules cm$^{-3}$ (Figure S8). These results suggest that the particle formation rate of GAS-involved SA-DMA system tends to become even larger at low temperature and high sulfuric acid concentration conditions.

**3.4 Enhancement effect of GAS on NPF**

To further evaluate the enhancing potential of GAS on SA-DMA driven NPF, $J_{\text{GAS-SA-DMA}}$ and $J_{\text{SA-DMA}}$ are compared by defining a ratio $r_{\text{GAS}}$, which stands for the ratio of cluster formation rate with GAS to that without GAS,

$$r_{\text{GAS}} = \frac{J([\text{GAS}] = x, [\text{SA}] = y, [\text{DMA}] = z)}{J([\text{SA}] = y, [\text{DMA}] = z)} \tag{4}$$

where [GAS], [SA], and [DMA] represent for monomer concentration of GAS, SA, and DMA, respectively. $r_{\text{GAS}}$ was calculated under the condition of [GAS] = $10^3$ to ~$10^5$ molecules cm$^{-3}$, [SA] = $10^4$-$10^7$ molecules cm$^{-3}$ and [DMA] = $10^8$ molecules cm$^{-3}$, which are the typical observed values in the atmosphere (Li et al., 2018;Riipinen et al., 2007;Kürten et al., 2014;Ge et al., 2011).

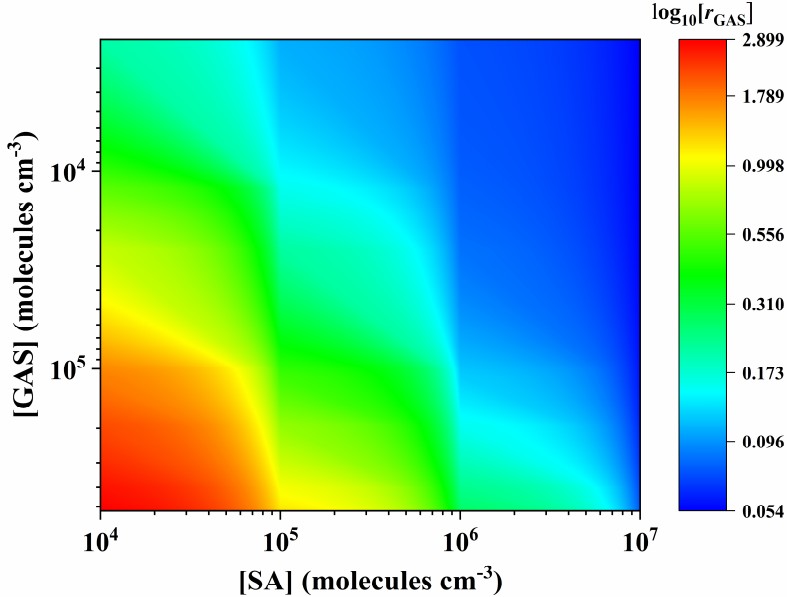

**Figure 4.** The cluster formation rates ratio ($r_{\text{GAS}}$) versus [SA] and [GAS] at [DMA] = $10^8$ molecules cm$^{-3}$ and 278 K. The color bars are values for $\log_{10}[r_{\text{GAS}}]$.

Figure 4 shows the enhancement strength $r_{\text{GAS}}$, which presents a dependence on [GAS] and [SA]. The $r_{\text{GAS}}$ increases with the increase of [GAS], and can reach as high as ~800 at 278 K with [GAS] = ~$10^5$ molecules cm$^{-3}$ and [SA] = ~$10^4$ molecules cm$^{-3}$, suggesting that the GAS can substantially enhance the formation rates of SA-DMA driven NPF, especially under relevant low [SA] atmospheric conditions. For instance, low concentrations of SA ~$10^4$ molecules cm$^{-3}$ together with relatively high NPF rates (5-9 cm$^{-3}$s$^{-1}$) were observed at Hyytiälä in Finland and Hohenpeissenberg in Germany (Mikkonen et al., 2011;Birmili et al., 2003;Maso et al., 2007). Since relatively high concentrations of hydroxy acids (259 ng m$^{-3}$) were also observed at the same period in nearby regions (Stieger et al., 2021), thereby the new particles may be mainly formed through ternary GAS-SA-DMA pathway rather than SA-DMA pathway in these scenarios. With [SA] increase, the $r_{\text{GAS}}$ becomes smaller as shown in Figure 4. This is likely due to the abundance advantage of SA compared to GAS in the concentration range we considered. When their concentrations are equal, GAS exhibits a certain enhancement effect. For example, $r_{\text{GAS}}$ still can reach ~3 at the condition of [GAS] = [SA] = $10^5$ molecules cm$^{-3}$, which indicates the nucleation capability between GAS and DMA is not

inferior to that of SA and DMA when the concentration of GAS and SA is at the same level. Previous studies have shown that the concentrations of organic precursors usually are several orders of magnitude higher than that of SA, when the enhancement strength $r$ is slightly larger than 1 (Li et al., 2017;Liu et al., 2021a;Zhang et al., 2018a). Such as, the enhancement effect of lactic acid (LA) on SA-DMA driven NPF at 260 K is approximately equals to 1, under the condition of [LA] = $10^{10}$ and [SA] = $10^5$ molecules cm$^{-3}$ (Li et al., 2017). Overall, it suggests that GAS has a non-negligible enhancement effect on the formation rate of SA-DMA binary nucleation system, in view of its binding property and the estimated equilibrium concentration in the atmosphere.

A series of kinetic simulations were further carried out by ACDC under different temperature and [DMA], so as to achieve a deeper understanding of enhancing potential of GAS. The influence of varying temperature (258 K, 278 K, and 298 K) and [DMA] ($10^7$, $10^8$, and $10^9$ molecules cm$^{-3}$) on cluster formation rates were shown in Figure 5. As shown in Figure 5a, the increasing trend of $J_{\text{GAS-SA-DMA}}$ are similar at different temperature, but the $J_{\text{GAS-SA-DMA}}$ markedly increases and reaches from $10^{-4}$ cm$^{-3}$s$^{-1}$ to 5 cm$^{-3}$s$^{-1}$ as temperature changes from 298 K to 258 K. The relatively low temperature can promote the cluster formation rate probably due to its effect on the decrease of evaporation rates for the related clusters, because the low temperature inhibits the endothermic thermodynamic process of cluster evaporation to some extent. The $J_{\text{GAS-SA-DMA}}$ also tend to increase 3 orders of magnitude with increasing [DMA] from $10^7$ molecules cm$^{-3}$ to $10^9$ molecules cm$^{-3}$ (Figure 5b). This is possibly owing to the fact that hydrogen bonding interaction increases between acidic molecules (GAS and SA) and the increased base molecules (DMA), which further results to the phenomenon of formation rate increase. These results demonstrate that the produced GAS from the chemical reaction of $SO_3$ with GA can speed up the SA-DMA nucleation even more dramatically under high [DMA] and at low temperature. It suggests that the enhancing potential of organosulfates on SA-DMA driven NPF deserves more attentions in highly amines polluted regions, especially with lower temperature, such as high mountains and cold polar areas.

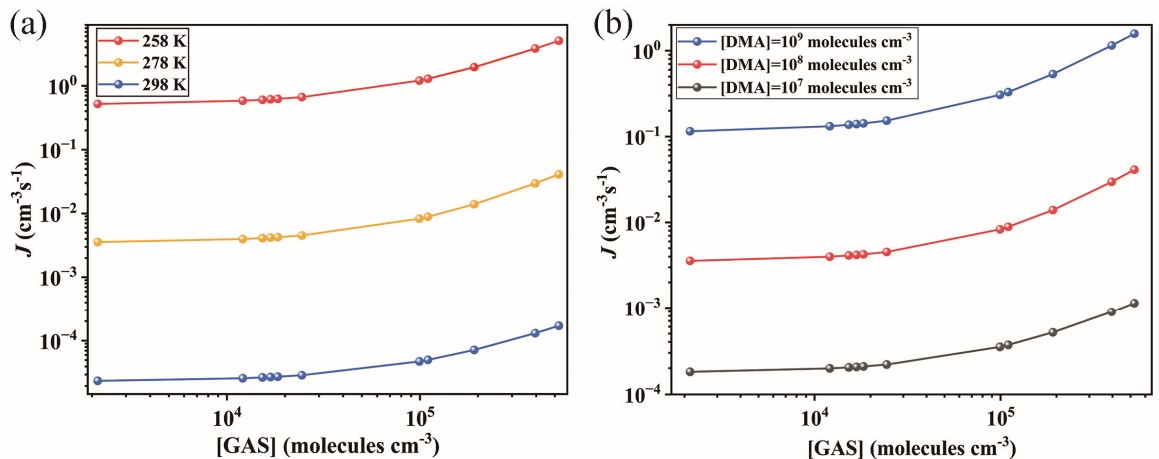

**Figure 5.** Simulated cluster formation rates $J_{\text{GAS-SA-DMA}}$ (cm$^{-3}$s$^{-1}$) (a) as a function of [GAS] and temperature at [DMA] = $10^8$ molecules cm$^{-3}$, [SA] = $10^5$ molecules cm$^{-3}$ and (b) as a function of [GAS] and [DMA] at 278 K and [SA] = $10^5$ molecules cm$^{-3}$

## 3.5 Cluster growth pathways

Figure 6 presents main growth pathways of GAS-SA-DMA-based clusters at different [GAS] ($10^4$-$10^6$ molecules cm$^{-3}$) with [SA] of $10^5$ molecules cm$^{-3}$ and [DMA] of $10^8$ molecules cm$^{-3}$ at 278 K, traced employing ACDC. When [GAS] is $10^4$ or $10^5$ molecules cm$^{-3}$ (Figure 6a, Figure 6b), the nucleation involves two primary pathways: 1) the pure SA-DMA nucleation pathway and 2) the GAS-SA-DMA nucleation pathway. In both pathways, the formation of $(SA)_1 \cdot (DMA)_1$ cluster is the first step from monomers of SA and DMA. The pure SA-DMA cluster (black arrows) grows by the stepwise addition of either SA or DMA, namely, each addition of an SA molecule is followed by one additional DMA molecule. In the case of GAS-SA-DMA nucleation pathway (green arrows), the initially formed GAS-involved cluster is $(GAS)_1 \cdot (SA)_1 \cdot (DMA)_1$, generated from GAS collision with the pre-existing $(SA)_1 \cdot (DMA)_1$ cluster. And then the $(GAS)_1 \cdot (SA)_1 \cdot (DMA)_1$ cluster grows via a base-stabilization mechanism, which is similar as the pure SA-DMA nucleation pathway. The ternary cluster growth path follows the sequence: $(GAS)_1 \cdot (SA)_1 \cdot (DMA)_1 \rightarrow (GAS)_1 \cdot (SA)_1 \cdot (DMA)_2 \rightarrow (GAS)_1 \cdot (SA)_2 \cdot (DMA)_2 \rightarrow (GAS)_1 \cdot (SA)_2 \cdot (DMA)_3 \rightarrow$ flux out. With the increase of [GAS] to $10^6$ molecules cm$^{-3}$ (Figure 6c), another alternative GAS-involved pathway appears. Of particular note, the third growth pathway formed from $(GAS)_1 \cdot (SA)_1 \cdot (DMA)_2$ cluster. Thereafter, the cluster growth proceeds via adding one GAS monomer to the $(GAS)_1 \cdot (SA)_1 \cdot (DMA)_2$ cluster, leading to the formation of $(GAS)_2 \cdot (SA)_1 \cdot (DMA)_2$ cluster. And then the $(GAS)_2 \cdot (DMA)_2$ clusters flux out after the subsequent evaporation of SA molecule. In this case, the contribution of GAS to the main cluster growth pathway increases from 3% (Figure 6a) to 78% (73% + 5%) (Figure 6c) with [GAS] increasing from $10^4$ to $10^6$ molecules cm$^{-3}$. The GAS molecules do participate in the growth of clusters and further flux out instead of evaporating from the existed clusters. Interestingly, in previous study about methyl hydrogen sulfate (MHS) (Li et al., 2018;Liu et al., 2019), which is mainly produced from atmospheric chemical reaction, the reported cluster pathway of MHS-SA-DMA system is very similar with that of GAS-SA-DMA system we observed. Therefore, the cluster growth pathway of GAS-SA-DMA system may represent a common feature for the mechanism of organosulfates participating in SA-DMA driven NPF.

The main growth pathways of GA-SA-DMA system were also investigated and compared with that of GAS-SA-DMA system (Figure S10). On the contrary, GA do not substantially conduce to the SA-DMA-based cluster growth, and the path only involve pure SA-DMA clusters at [GA] = $10^9$ molecules cm$^{-3}$ (Figure S10b). Till [GA] increases to $10^{10}$ molecules cm$^{-3}$ (Figure S10c), GA can indirectly participate in the cluster growth, acting as a "mediate bridge" and finally evaporating. This is consistent with the probed "mediate bridge" mechanism of GA-SA-NH$_3$ system (Zhang et al., 2017). The GA molecule in GA-SA-DMA system initially participates in the cluster growth and then evaporates from the newly formed clusters mainly on account of high evaporation rates of GA containing clusters, which ranges from 20 to $10^6$ s$^{-1}$ for $(GA)_1(SA)_2(DMA)_3$, $(GA)_2(SA)_1(DMA)_3$, and $(GA)_3(DMA)_3$ clusters. The growth pathways of GAS-SA-DMA and GA-SA-DMA systems are obviously different. It can be concluded that GAS is an important "participator" rather than GA just as a "mediate bridge", when participating in the SA-DMA nucleation. This difference could mainly result from the higher

stability of the $(GAS)_1(SA)_2(DMA)_3$ and $(GAS)_3(DMA)_3$ clusters compared to those of $(GA)_1(SA)_2(DMA)_3$, $(GA)_2(SA)_1(DMA)_3$ and $(GA)_3(DMA)_3$ clusters with high evaporation rates (in all cases larger than $10^1$ s$^{-1}$) at the studied precursor concentrations (Figure 2, Figure S7 and Table S6), thereby the GA-involved clusters evaporating into smaller clusters.

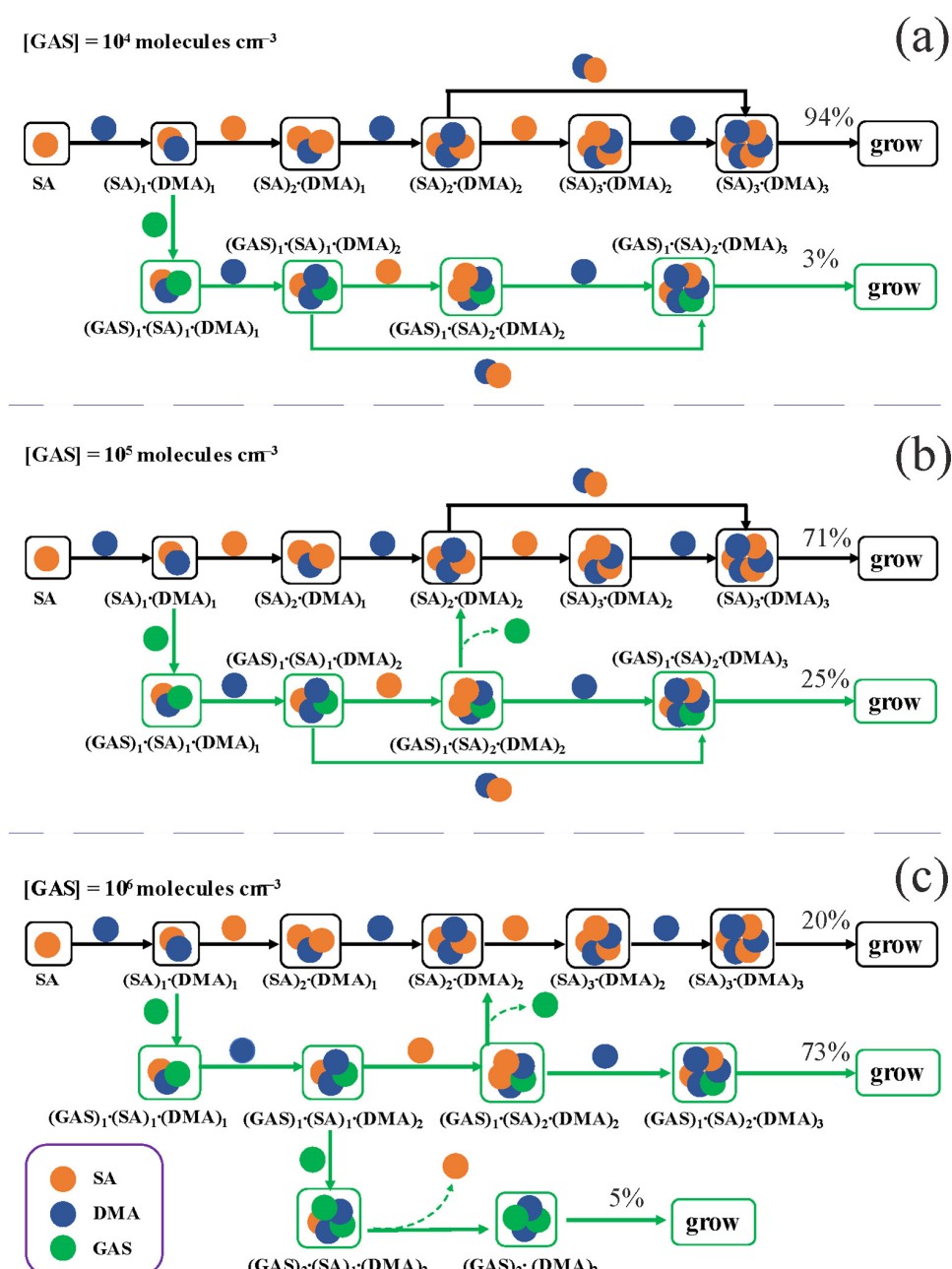

**Figure 6.** Main cluster growth pathway of GAS-SA-DMA nucleation system at 278 K, [DMA] = $10^8$, [SA] = $10^5$, and (a) [GAS] = $10^4$, (b) [GAS] = $10^5$, (c) [GAS] = $10^6$ molecules cm$^{-3}$. The black and green arrows refer to the pathways of SA-DMA and GAS-SA-DMA, respectively. For clarity, other pathways that contributes less than 5% to the cluster growing out of the studied system are not shown.

The cluster growth pathways of the GAS-SA-DMA system at different temperature (258 K, 278 K and 298 K) were also compared (Figure S11). The proportion of GAS-involved path increases with the decreasing temperature. At 258 K, the $(SA)_2(DMA)_2$ cluster can directly form the $(GAS)_1(SA)_2(DMA)_2$ cluster through the addition of one GAS molecule. In that case, SA-DMA-based clusters can even assist the growth of clusters containing GAS from the 2:2 size, contributing to another growth pathway to form large GAS-involved clusters. The founded alternative pathway emerges from SA-DMA-based cluster

to form $(GAS)_1 \cdot (SA)_2 \cdot (DMA)_2$, leading to the proportion of GAS-involved path increase up to 53%. This phenomenon may

be because low temperature could efficiently promote the thermodynamical stabilization of ternary clusters. Such a positive

correlation of cluster stability with low temperature has also been suggested to be a common feature in studies of other ternary

systems containing acidic compounds, e.g., MSA, MSIA, NA and so on (Liu et al., 2021b;Ning and Zhang, 2022;Ning et al.,

2020).

**3.6 Comparison with observations**

Intense NPF events as well as emission of a suite of various nucleation precursors have been observed at the summit of Mt.

Tai in China recent years (Mochizuki et al., 2017;Lv et al., 2018). The gaseous concentration of GA reaches 343 ng m$^{-3}$ during

more field burning influenced periods at Mt. Tai (Mochizuki et al., 2017). And the formation rates of 3nm particles lied in the

range of 0.82-25.04 cm$^{-3}$ s$^{-1}$ (Lv et al., 2018). The condensation sink (CS) (with the average of $1.4 \times 10^{-2}$ s$^{-1}$) and air temperature

were found to be lower, whereas the concentration of SO$_2$ was higher on NPF days than that on non-NPF days. A strong

correlation existed between the continental air mass passing through polluted regions and NPF, which was partly because of

the higher SO$_2$ concentration, indicating that SA was an important precursor on Mt. Tai. However, the pure SA-base nucleation

could not fully explain the observed cluster formation rates ($J_{obs}$) attributed to the deficiency of SA concentration. Given the

fact that GA has only a slight influence on the nucleation and growth processes of atmospheric clusters, the reaction between

GA and SO$_3$ may provide a secondary source of the potential precursor since high concentrations of sulfur oxides being

detected. Interestingly, in this work its product, GAS, was identified to have the ability to stabilize SA-DMA-based clusters,

and be able to speed up SA-DMA nucleation obviously. Hence, coupled the published observational evidence at Mt. Tai in

China (Mochizuki et al., 2017;Lv et al., 2018) and our aforementioned theoretical analysis, it can be supposed that these NPF

events involve GAS.

In Figure 7, we plotted the cluster formation rates for pathway SA-DMA, GA-SA-DMA and GAS-SA-DMA, individually.

Note that the calculated cluster formation rates via ACDC simulation in this work are at a cluster size about ~ 1.3 nm. The

observed cluster formation rates ($J_3$) at Mt. Tai were measured at 3 nm. According to the revised Kerminen−Kulmala equation

(Anttila et al., 2010; Lehtinen et al., 2007), the cluster formation rate $J_3 \approx 0.5\ J_{1.3}$ (see details in the SI). Hence, $J_{3nm-(SA-DMA)}$,

$J_{3nm-(GA-SA-DMA)}$ and $J_{3nm-(GAS-SA-DMA)}$ presented in Figure 7 were calculated from 0.5 times their associated cluster formation

rates ($J_{SA-DMA}$, $J_{GA-SA-DMA}$ and $J_{GAS-SA-DMA}$, respectively) obtained via the ACDC simulations. Here we can see the needed

concentration of SA for binary SA-DMA (dotted lines) is clearly higher than those of for ternary GA-SA-DMA (dashed lines)

and GAS-SA-DMA (solid lines) system at the condition of the same formation rates. The needed [SA] for GAS-SA-DMA

system is obviously lower than that of for GA-SA-DMA, and markedly reduces with [GAS] increase. In contrast, although

[GA] is higher than [GAS] in Figure 7, the variation of needed [SA] for GA-SA-DMA system is minor with the increase of

[GA]. These results imply that the influence of GAS on the SA-DMA system is stronger than that of GA, and the ternary GAS-

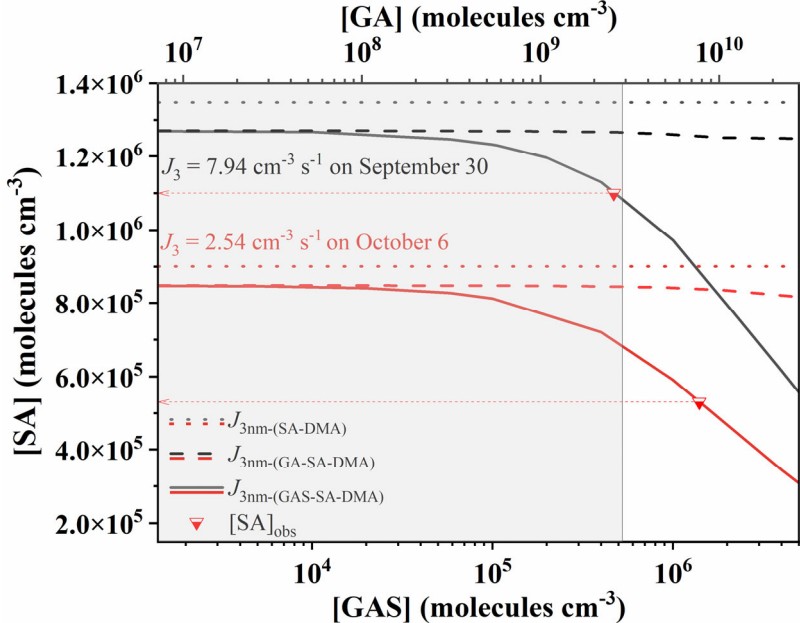

**Figure 7.** Required atmospheric concentrations of gas-phase precursors for pathways SA-DMA, GA-SA-DMA, and GAS-SA-DMA to reach the observed cluster formation rates ($J_3$) on 30 September 2014 (black lines) and 6 October 2014 (red lines) observed at Mt. Tai in China. [DMA] was set to be $10^8$ molecules $cm^{-3}$. Dotted red lines pointing from inverted triangles to arrows represent the observed [SA] at September 30 and 6 October 2014, respectively. The shaded area represents the globally observed [GA] and corresponding [GAS]. Simulated $J_{3nm-(SA-DMA)}$, $J_{3nm-(GA-SA-DMA)}$, and $J_{3nm-(GAS-SA-DMA)}$ are represented by dotted lines, dashed lines, and solid lines, individually. Observation data of [GA] and particle formation rates ($J_3$) come from ref. (Mochizuki et al., 2017) and ref. (Lv et al., 2018), respectively.

SA-DMA mechanism provides a new pathway for the NPF events with the condition of relatively low [SA] observed at Mt. Tai. The shaded area shown in Figure 7 represents the globally observed [GA] as well as corresponding [GAS]. The [SA] at Mt. Tai on September 30, 2014 is observed at $1.09 \times 10^6$ molecules $cm^{-3}$ (the top red line pointing from an inverted triangle to the left arrow). If the new particles at Mt. Tai on September 30, 2014 are presumed to be produced from the pure SA-DMA system with the typical [DMA] of $10^8$ molecules $cm^{-3}$, the concentration of SA around $\sim 1.35 \times 10^6$ molecules $cm^{-3}$ is needed (the black dotted line), which is quite higher than the observed [SA]. To reach the observed $J_{September\ 30}$ (7.94 $cm^{-3}s^{-1}$), the required [GAS] relevant to the observed [SA] on September 30, 2014, is $\sim 4.70 \times 10^5$ molecules $cm^{-3}$, in the shaded area as shown in Figure 7. This indicates that the ternary GAS-SA-DMA nucleation mechanism corresponds well with the observed records of [SA] and NPF events. As for the GA-SA-DMA pathway, the required [GA] and [SA] are presented by black/red dashed lines. It is very clear that the GA-SA-DMA ternary system is not sufficient enough to unravel the observed NPF at Mt. Tai, for that the corresponding [GA] to the observed [SA] is beyond the boundary of shaded area. For another example, if the new particles on October 6, 2014 are assumed to be generated from pure SA-DMA system, the required [SA] is estimated to be $\sim 9.01 \times 10^5$ molecules $cm^{-3}$ (the red dotted line), which is also too high for the observed [SA] ($5.3 \times 10^5$ molecules $cm^{-3}$, the bottom red line pointing from inverted triangle to the left arrow). Although GAS can speed up the SA-DMA driven NPF (the red line), to reach the observed $J_{October\ 6}$ (2.54 $cm^{-3}s^{-1}$), a fairly high concentration of GAS is required, which is out of the shaded area. This suggests that there may be other potential enhanced mechanisms for atmospheric new particle formation,

which require more in-depth researches. These results and analyses suggest the GAS-SA-DMA nucleation mechanism is possible to explain the field observation on atmospheric SA-involved particles at Mt. Tai, while the binary SA-DMA nucleation

is incompatible with the observed new particle formation rates. Therefore, it can be concluded that the GAS produced from the chemical reaction of GA and $SO_3$ could play an important role to speed up the SA-DMA driven NPF events at Mt. Tai. In the light of the deficiency of field observation of GAS in the gas phase, the further detection of GAS is still needed.

## 4 Atmospheric Implications

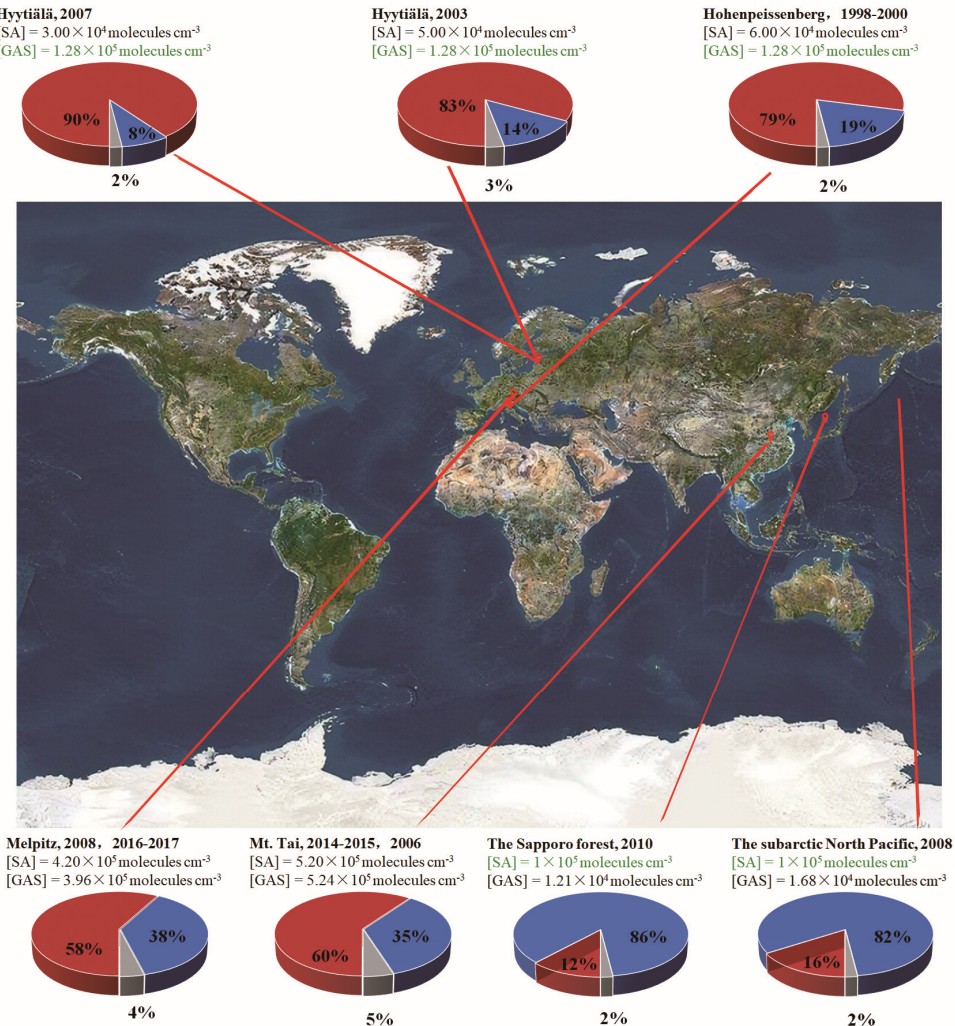

**Figure 8.** The branch ratio of the GAS-SA-DMA (red pie) and SA-DMA (blue pie) growth pathways based on field data in different regions with different [GAS]. The data recorded in black are from field observation and those of green are set to be a median in this study. [DMA]=$10^8$ molecules cm$^{-3}$. The map is from © Google Maps (https://www.google.com/maps).

This study reveals that the reaction of GA and $SO_3$ can generate a certain concentration of GAS as a potential atmospheric nucleation precursor, and the GAS is able to intensely speed up the SA-DMA nucleation. Therefore, GA in the atmosphere

can consume part of $SO_3$, which may lead to the observed relatively low SA concentration, but its product GAS can significantly enhance the SA-DMA-driven NPF under such conditions. Considering the high atmospheric gas-phase concentrations of GA and particle-phase GAS detected in diverse environments at regions worldwide, including Finnish forest,

German rural, Japanese forest, marine atmosphere of North Pacific and so on (Ehn et al., 2010;Miyazaki et al., 2014;Mochizuki et al., 2019;Mochizuki et al., 2017;Stieger et al., 2021), GAS could be an important contributor to SA-DMA driven NPF in locations with high GA concentration and relatively low SA concentration. To further evaluate the implication of GAS on the SA-DMA nucleation in the atmosphere, the specific contribution of SA-DMA cluster growth paths with/without GAS to NPF were calculated under the ambient conditions in corresponding regions (Figure 8). The branch ratios of the major flux out were investigated under varying [GAS] ($1.21 \times 10^4$ molecules cm$^{-3}$ ~ $5.24 \times 10^5$ molecules cm$^{-3}$) and [SA] ($3 \times 10^4$ molecules cm$^{-3}$ ~ $5.20 \times 10^5$ molecules cm$^{-3}$) at 278 K, which were basically from the field observations, including Mt. Tai (36.26° N, 117.11° E), Sapporo (42°59′N, 141°23′E), Melpitz (51°32′N, 12°56′E), Hyytiälä (61°51′N, 24°17′E), Hohenpeissenberg (47°48′N, 11°00′E) and the subarctic North Pacific.(Stieger et al., 2021;Mochizuki et al., 2017;Mochizuki et al., 2019;Miyazaki et al., 2014;Mikkonen et al., 2011) The concentrations of GAS/SA recorded in green as well as that of DMA are set to be a median in this study ($1 \times 10^5$ molecules cm$^{-3}$ for [SA], $1.28 \times 10^5$ molecules cm$^{-3}$ for [GAS], and $1 \times 10^8$ molecules cm$^{-3}$ for [DMA]). As presented in Figure 8, the branch ratio of the flux out is very sensitive to the [GAS]. In the high [GAS] regions, such as Melpitz ($3.96 \times 10^5$ molecules cm$^{-3}$) and Mt. Tai ($5.24 \times 10^5$ molecules cm$^{-3}$), the contributions of GAS-SA-DMA growth pathways (red pie in Figure 8) are dominant in their NPF. For regions with relatively low [GAS], e.g., Sapporo and subarctic North Pacific, the contributions of GAS-involved clustering pathways are 12% and 16%, respectively. Especially, in the region with low SA abundance, like Hyytiälä, the nucleation was also identified to be dominated by GAS-SA-DMA path, resulting in around 80% to the cluster formation. This implies that the influence of GAS in regions with relatively low SA concentration is also prominent. Note that the GAS concentration we discussed in this work are estimated from limited observational data of SO$_3$ and GA in the atmosphere. The actual atmospherics concentration of GAS still requires a large number of field observations to achieve more in-depth research.

This study found that low temperature and high DMA concentration are both favourable conditions for the higher enhancing potential of GAS to the SA-DMA nucleation. Therefore, in cold areas, the contribution of GAS to NPF deserves more attention, especially under the polluted conditions of high DMA abundance. The identified nucleation mechanism of GAS-SA-DMA system, in which GAS playing a participator role can not only promote the initial nucleation but also participate in the subsequent nucleation processes, also provides a feasible potential source for organosulfate in aerosol. If ignoring the contribution of organosulfate produced from chemical reaction to the NPF, the risk of hydroxy acids emissions and the sources of organic aerosols may be underestimated to some extent. Recently, more and more field observation data of hydroxy acids in diverse environments worldwide have been reported (Chen et al., 2021;Mochizuki et al., 2016;Duncan et al., 2019). The identified reaction mechanism in this study appears to be generalizable to evaluate the role of these acids, such as lactic acid, and their derivates in atmospheric NPF. To the best of the authors' knowledge, the roles of hydroxy acids as well as their derivates in atmospheric NPFs have not been systematically reported before. The current findings imply the necessities to further study the NPFs affected via chemical reactions of organic acids in the atmosphere. Including this new organosulfate

chemistry into the existing atmospheric models will improve the quantitative modelling of the effect of organic acids on atmospheric aerosol formation. Lastly, organosulfates produced from secondary source, like gas-phase chemical reactions, are deserving of further monitoring and evaluating.

**ACKNOWLEDGMENTS**

This work was supported by the National Natural Science Foundation of China (No. 21976061), GuangDong Basic and
500 Applied Basic Research Foundation (No. 2022A1515010591).

**Supporting Information**

Computational details for the concentration of GAS as well as GASA, judging of GA-SA-DMA, GAS-SA-DMA cluster, and GASA-SA-DMA cluster stability and boundary conditions, variables settings of ACDC, catalytic effect of GA on $SO_3$-$H_2O$ reaction, conformations of GA-SA-DMA, GAS-SA-DMA clusters and GASA-SA-DMA clusters, evaporation coefficients for
all evaporation pathways of clusters investigated in this work, effect of [SA] on GA-SA-DMA, GAS-SA-DMA clusters, and GASA-SA-DMA cluster formation rates, the main cluster growth pathways of GAS-SA-DMA system in comparison with that of GA-SA-DMA system, details for thermodynamics information for the formation of GA-SA-DMA, GAS-SA-DMA clusters, and GASA-SA-DMA clusters.

**Corresponding Author**

Shi Yin - *MOE & Guangdong Province Key Laboratory of Laser Life Science & Institute of Laser Life Science, Guangzhou Key Laboratory of Spectral Analysis and Functional Probes, College of Biophotonics, South China Normal University, Guangzhou 510631, P. R. China*; E-mail: yinshi@m.scnu.edu.cn

**Data availability.** Data from this research are not publicly available. Interested researchers can contact the corresponding author of this article.

**Author contributions.** Conceptualization of the research goals, development of the methodology, and construction of the models was completed by XZ and ST under the supervision of SY. YL assisted in data analyses. XZ prepared the original draft, which was subsequently reviewed and edited by all co-authors.

**Competing interests.** The authors declare that they have no conflict of interest.

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
