# Peer review of "Organosulfate Produced from Consumption of SO3 Speeds up Sulfuric Acid-Dimethylamine Atmospheric Nucleation"

_EGUsphere, 2023_

## Author Response (AR1)

Shi Yin

Associate Professor

College of Biophotonics,

South China Normal University,

Guangzhou 510631, P. R. China

Email: yinshi@m.scnu.edu.cn

Dear Prof. Jason Surratt,

Many thanks for your attention to our manuscript. We have revised our manuscript (egusphere-2023-1649) "*Organosulfate Produced from Consumption of SO₃ Speeds up Sulfuric Acid-Dimethylamine Atmospheric Nucleation*", according to reviewers' comments. We appreciate the reviewer's suggestions and comments, and have addressed all their issues and questions. We list below the corresponding changes and explanations we have added to the text, as requested by the reviewers. A marked copy of the manuscript with yellow highlight annotation for our changes were also uploaded for the Editor and Reviewers convenience.

Best regards,

Shi Yin

Shi Yin

**Response to referee #1:**

Referee's Comments to Author:

*The manuscript entitled 'Organosulfate produced from consumption of SO3 speeds up sulfuric acid-dimethylamine atmospheric nucleation' by Zhang et al. presents a detailed theory of the potential role of gas-phase glycolic acid (and its organosulfate derivatives) in new particle formation (NPF). The authors rely on quantum chemical calculations to define the most reasonable pathway and formation product of the reaction between glycolic acid and SO3, leading to organosulfates. Also, kinetic modeling is used to understand the most efficient clustering with dimethyl amines. Their findings are backed up with atmospheric observations from Mt. Tai in China where intense NPF events have been observed. The authors also report observations from other locations around the world where the role of glycolic acid in NPF might be important but has been not evaluated earlier. First of all, I would like to acknowledge the authors for a very detailed, easy to follow manuscript. As mentioned by the authors, there are very few studies which tackle the role of organic acids in NPF, mainly due to the scarcity of their measurements. The results presented in this manuscript improve our understanding of the formation pathway of organosulfates from glycolic acid and sulfate and the role of the most stable organosulfate product in clustering with dimethyl amine and its role in NPF. The paper calls for an inclusion of organosulfate chemistry in global models when evaluating NPF and for more studies tackling hydroxyl acids in the gas phase from an observational and measurement point of view. The paper is well written and the finding are sound, I recommend publication after tackling the suggestions listed below.*

*General comments:*

*1.  Could the authors comment on the role of relative humidity (availability of H2O clusters) and how these affect the formation of GAS and GASA. Please see results by Tsona et al. (https://doi.org/10.1016/j.atmosenv.2019.116921)*

Author reply:

Thanks for the referee's question and suggestions. In order to comment the role of relative humidity (availability of $H_2O$ clusters) and how these affect the formation of GAS and GASA, we further explored hydration process of glycolic acid (GA) in the gas phase, according to the method reported by Tsona et al. (Tsona and Du, 2019) at tropospheric temperatures (258 K, 278 K, 298 K) and ambient pressure. The lowest-energy structure of clusters GA-$n$($H_2O$) ($n$ = 0 - 3) (Figure S2), equilibrium distribution of GA hydrates (Figure S3), thermodynamic data of the stepwise hydration of GA (Table S2), and relative equilibrium abundance of GA hydrates at different degrees of humidity (Table S3) were obtained and added to the revised supporting information of the manuscript. Following discussions were added to page S7 in our modified supporting information materials.

"As shown in Table S2, the first water addition to GA at 298 K, with a Gibbs free energy change of -0.71 kcal mol$^{-1}$, are more favorable compared to the second and the third water additions, having free energy changes of 2.98 and 0.18 kcal mol$^{-1}$, respectively. Additionally, the energies decrease as the temperature decreases, and the energy of first water addition reaches to -1.96 kcal mol$^{-1}$ at 258 K. Although the abundances of hydrated GA clusters, including mono-, di- and tri-hydrates, are slightly increased with increasing RH (relative humidity) (Figure S3 and Table S3), the hydration of GA is still weak at high RH. For example, the relative equilibrium abundance of GA hydrates is less than 7% at RH = 90% and 298 K. Therefore, the relative abundance of unhydrated GA is dominant at varying temperature and RH considered."

Following sentences were also added to line 175 in page 6 in our modified manuscript.

"These results indicate that both reaction pathways for GA + SO$_3$ are favorable with the catalysis of H$_2$O to generate GAS and GASA, respectively. Therefore, as the relative humidity (RH) increases, it should be conducive to the formation of GAS and GASA. The abundances of hydrated GA clusters GA-$n$(H$_2$O) ($n$ = 0 - 3) were also calculated at different RH (Figure S3 and Table S3). The relative equilibrium abundance of GA hydrates is less than 7% at RH = 90% and 298 K. Since the hydration of GA is weak, the effect of the hydrated GA clusters to the formation of GAS and GASA is not further considered."

[Figure]

**Figure S2** Lowest energy geometries of (GA)(H$_2$O)$_n$, optimized with the M06-2X/6-311++G(3df,3pd) method. The color coding is red for oxygen, grey for carbon, and white for hydrogen. The number of water molecules increases from (a) where $n$ = 0 to (d) where $n$ = 3.

[Figure]

**Figure S3** Equilibrium distribution of glycolic acid hydrates at different degrees of humidity (RH = 10%, 50% and 90%) and different temperatures (258 K, 278 K, and 298 K).

**Table S2** Thermodynamic data (in kcal mol$^{-1}$) of the stepwise hydration of glycolic acid (GA) calculated by the M06-2X/6-311++G(3df,3pd) method.

| n | $\Delta G$ | | |
| --- | --- | --- | --- |
| | 258 K | 278 K | 298 K |
| | $(GA)(H_2O)_{n-1} + H_2O \leftrightarrow (GA)(H_2O)_n$ | | |
| 1 | -1.96 | -1.33 | -0.71 |
| 2 | 1.77 | 2.38 | 2.98 |
| 3 | -1.07 | -0.44 | 0.17 |

**Table S3** Relative equilibrium abundance of GA hydrates at different degrees of humidity (RH = 10%, 20%, 40%, 50%, 90% and 100%) and different temperatures (258 K, 278 K, and 298 K)

**258 K**

| RH | 10% | 20% | 40% | 50% | 90% | 100% |
| --- | --- | --- | --- | --- | --- | --- |
| n=0 | 0.98970000 | 0.97970000 | 0.96020000 | 0.95070000 | 0.91460000 | 0.90600000 |
| n=1 | 0.01026766 | 0.02032661 | 0.03984325 | 0.04931278 | 0.08539284 | 0.09398853 |

| n=2 | 0.00000006 | 0.00000025 | 0.00000098 | 0.00000151 | 0.00000472 | 0.00000577 |
| --- | --- | --- | --- | --- | --- | --- |
| n=3 | 0.00000002 | 0.00000018 | 0.00000139 | 0.00000268 | 0.00001504 | 0.00002044 |

**278 K**

| RH | 10% | 20% | 40% | 50% | 90% | 100% |
| --- | --- | --- | --- | --- | --- | --- |
| n=0 | 0.99050000 | 0.98120000 | 0.96310000 | 0.95430000 | 0.92070000 | 0.91260000 |
| n=1 | 0.00948502 | 0.01879180 | 0.03689036 | 0.04569154 | 0.07934438 | 0.08738994 |
| n=2 | 0.00000001 | 0.00000004 | 0.00000015 | 0.00000023 | 0.00000073 | 0.00000089 |
| n=3 | 0.00000000 | 0.00000001 | 0.00000009 | 0.00000017 | 0.00000095 | 0.00000130 |

**298 K**

| RH | 10% | 20% | 40% | 50% | 90% | 100% |
| --- | --- | --- | --- | --- | --- | --- |
| n=0 | 0.99260000 | 0.98530000 | 0.97100000 | 0.96400000 | 0.93700000 | 0.93050000 |
| n=1 | 0.00741927 | 0.01472926 | 0.02903091 | 0.03602716 | 0.06303219 | 0.06954867 |
| n=2 | 0.00000000 | 0.00000000 | 0.00000001 | 0.00000002 | 0.00000006 | 0.00000008 |
| n=3 | 0.00000000 | 0.00000000 | 0.00000000 | 0.00000000 | 0.00000002 | 0.00000003 |

**Reference**

Tsona, N. T., and Du, L.: Hydration of glycolic acid sulfate and lactic acid sulfate: Atmospheric implications, Atmospheric Environment, 216, 116921, https://doi.org/10.1016/j.atmosenv.2019.116921, 2019.

*2.    Could the authors comment on the role of coagulation sink? Eg. Line 134, it is a reasonable assumption to use the values from Hyytiälä for Mt. Tai for example?*

Author reply:

Thanks for the referee's question. According to the reviewer's suggestion, the following detail descriptions about coagulation sink were clarified and modified in line 155 page 6 in our revised manuscript. "A constant coagulation sink of $2.6 \times 10^{-3}$ s$^{-1}$ was applied to account for scavenging by larger particles. The simulations were mainly run at 278 K, with additional runs at 258 K and 298 K to investigate the influence of temperature. These conditions correspond to a typical sink value and temperature in the boreal forest environment (Olenius et al., 2013; Maso et al., 2008)." A constant

coagulation sink was applied in the cluster distribution dynamics simulations to account for scavenging by larger particles. The coagulation sink used in this work is the major loss by particles in the assumed atmospheric conditions. Mt. Tai is located in northern China, and its landscape is dominated by forests. Unfortunately, we did not find the measurement coagulation sink report in Mt. Tai. We chose a constant coagulation sink coefficient of $2.6 \times 10^{-3}$ s$^{-1}$, which is the median condensation sink coefficient of sulfuric acid vapor on pre-existing aerosol particles, based on measurements in the boreal forest environment in Hyytiälä, Finland (Maso et al., 2007). On the other hand, the cluster size dependent coagulation sink coefficient did not have a significant effect on the steady-state cluster concentrations, according to the parametrized formula from Kulmala et al. (Kulmala et al., 2001; Kulmala and Wagner, 2001). Additionally, this coagulation sink value is widely used for a typical sink for molecular sized clusters in continental background areas (Paasonen et al., 2012), and taking into account external losses of organic compound-sulfuric acid-dimethylamine cluster system (Li et al., 2017). Therefore, we think it is a reasonable assumption to use this coagulation sink value in this work.

**Reference**

Maso, M. D., Sogacheva, L., Aalto, P. P., Riipinen, I., Komppula, M., Tunved, P., Korhonen, L., Suur-Uski, V., Hirsikko, A., KurtéN, T., Kerminen, V.-M., Lihavainen, H., Viisanen, Y., Hansson, H.-C., and Kulmala, M.: Aerosol size distribution measurements at four Nordic field stations: identification, analysis and trajectory analysis of new particle formation bursts, 660 Tellus B, 59, 350-361, https://doi.org/10.1111/j.1600-0889.2007.00267.x, 2007.

Kulmala, M. and Wagner, P. E.: Mass accommodation and uptake coefficients – a quantitative comparison , J. Aerosol Sci., 32, 833–841, https://doi.org/10.1016/S0021-8502(00)00116-6, 2001.

Kulmala, M., Maso, M. D., Mäkelä, J. M., Pirjola, L., Väkevä, M., Aalto, P., Miikkulainen, P., Hämeri, K., And O'dowd, C. D.: On the formation, growth and composition of nucleation mode particles, Tellus B, 53, 479-490, https://doi.org/10.1034/j.1600-0889.2001.530411.x, 2001.

Li, H., Kupiainen-Määttä, O., Zhang, H., Zhang, X., and Ge, M.: A molecular-scale study on the role of lactic acid in new particle formation: Influence of relative humidity and temperature, Atmospheric Environment, 166, 479-487, https://doi.org/10.1016/j.atmosenv.2017.07.039, 2017.

Paasonen, P., Olenius, T., Kupiainen, O., Kurtén, T., Petäjä, T., Birmili, W., Hamed, A., Hu, M., Huey, L. G., Plass-Duelmer, C., Smith, J. N., Wiedensohler, A., Loukonen, V., McGrath, M. J., Ortega, I. K., Laaksonen, A., Vehkamäki, H., Kerminen,

V. M., and Kulmala, M.: On the formation of sulphuric acid - amine clusters in varying atmospheric conditions and its influence on atmospheric new particle formation, Atmos. Chem. Phys., 12, 9113-9133, 10.5194/acp-12-9113-2012, 2012.

*3.    The products GAS and GASA are better introduced in the main text. Could the authors move page S3 from the supplementary information to the main text in the methods section?*

Author reply:

We appreciated the referee's suggestion. According to the suggestion, we moved the page S3 to the methods section (2.3 The concentration of glycolic acid sulfate (GAS) and glycolic acid sulfuric anhydride (GASA) ) of main text in page 4 in our revised manuscript.

*4.    References to studies who tackled the role of organosulfates and other similar organic acids in NPF are missing e.g. Katz et al. (https://doi.org/10.5194/acp-23-5567-2023) and Zhang et al. (https://doi.org/10.5194/acp-22-2639-2022)*

Author reply:

Thanks for the referee's comments. According to the suggestion, we have read articles, introducing the role of organosulfates and other similar organic acids in NPF by Katz et al. and Zhang et al. The following sentences and references were added to line 37 and line 51 page 2 in the Introduction part of the revised manuscript, respectively.

"Organosulfates have been identified as the most abundant class of organosulfur compounds, accounting for 5-30% of the organic mass fraction in atmospheric particles (Brüggemann et al., 2017; Tolocka and Turpin, 2012; Shakya and Peltier, 2015; Froyd et al., 2010; Mutzel et al., 2015; Glasius et al., 2018). Katz et al. measured the presence of organosulfates and identified its importance to new particle formation (Katz et al., 2023)."

"Organic acids, which are frequently observed in the atmosphere, have been expected to participate in the process of atmospheric nucleation, with a focus on the thermochemical properties of clusters between organic acids and common atmospheric nucleation precursors (Zhang et al., 2022)."

**Reference**

Brüggemann, M., Poulain, L., Held, A., Stelzer, T., Zuth, C., Richters, S., Mutzel, A., van Pinxteren, D., Iinuma, Y., Katkevica, S., Rabe, R., Herrmann, H., and Hoffmann, T.: Real-time detection of highly oxidized organosulfates and BSOA

marker compounds during the F-BEACh 2014 field study, Atmos. Chem. Phys., 17, 1453-1469, https://doi.org/10.5194/acp-17-1453-2017, 2017.

Tolocka, M. P., and Turpin, B.: Contribution of organosulfur compounds to organic aerosol mass, Environ. Sci. Technol., 46, 7978-7983, https://doi.org/10.1021/es300651v, 2012.

Shakya, K. M., and Peltier, R. E.: Non-sulfate sulfur in fine aerosols across the United States: Insight for organosulfate prevalence, Atmos. Environ., 100, 159-166, https://doi.org/10.1016/j.atmosenv.2014.10.058, 2015.

Froyd, K. D., Murphy, S. M., Murphy, D. M., de Gouw, J. A., Eddingsaas, N. C., and Wennberg, P. O.: Contribution of isoprene-derived organosulfates to free tropospheric aerosol mass, Proc. Natl. Acad. Sci. U.S.A., 107, 21360, https://doi.org/10.1073/pnas.1012561107, 2010.

Mutzel, A., Poulain, L., Berndt, T., Iinuma, Y., Rodigast, M., Böge, O., Richters, S., Spindler, G., Sipilä, M., Jokinen, T., Kulmala, M., and Herrmann, H.: Highly Oxidized Multifunctional Organic Compounds Observed in Tropospheric Particles: A Field and Laboratory Study, Environ. Sci. Technol., 49, 7754-7761, https://doi.org/10.1021/acs.est.5b00885, 2015.

Glasius, M., Hansen, A. M. K., Claeys, M., Henzing, J. S., Jedynska, A. D., Kasper-Giebl, A., Kistler, M., Kristensen, K., Martinsson, J., Maenhaut, W., Nøjgaard, J. K., Spindler, G., Stenström, K. E., Swietlicki, E., Szidat, S., Simpson, D., and Yttri, K. E.: Composition and sources of carbonaceous aerosols in Northern Europe during winter, Atmos. Environ., 173, 127-141, https://doi.org/10.1016/j.atmosenv.2017.11.005, 2018.

Katz, D. J., Abdelhamid, A., Stark, H., Canagaratna, M. R., Worsnop, D. R., and Browne, E. C.: Chemical identification of new particle formation and growth precursors through positive matrix factorization of ambient ion measurements, Atmos. Chem. Phys., 23, 5567–5585, https://doi.org/10.5194/acp-23-5567-2023, 2023.

Zhang, R., Shen, J., Xie, H. B., Chen, J., and Elm, J.: The Role of Organic Acids in New Particle Formation from Methanesulfonic Acid and Methylamine, Atmos. Chem. Phys. Discuss., 2021, 1-18, https://doi.org/10.5194/acp-2021-831, 2022.

*Technical comments:*
*1.    In figure 3, could the authors write in the figure caption that the GA-SA-DMA lines are the same in a and b, but the y scale is different?*

Author reply:

Thanks for the referee's suggestion. In our modified manuscript, we made a detailed description in the figure caption of Figure 3 that the GA-SA-DMA lines are same in a and b, but the Y-axis scale is

different.

[Figure]

Figure 3. Simulated cluster formation rates $J$ (cm$^{-3}$s$^{-1}$) as a function of monomer concentrations ([GA], [GAS], and [GASA], respectively) at (a) (b) 278 K and (c) (d) 258 K under the condition of [DMA] = $10^8$ molecules cm$^{-3}$ and [SA] = $10^5$ molecules cm$^{-3}$. Note that the simulated $J_{\text{GA-SA-DMA}}$ are the same data, but the Y-axis scale are different at (a) (b) and (c) (d), individually.

*2. References format needs to be checked. In some cases, e.g. line 149, 150 the citation is starting the sentence.*

Author reply:

Thanks for the referee's comment. According to the comment, we made an effort to check and correct all reference format in our modified manuscript. The sentence of line 149,150 "With high abundance (~$10^{17}$ cm$^{-3}$) being detected in the troposphere,(Huang et al., 2015) $H_2O$ has been reported to effectively act as a catalyst in chemical reactions.(Liu et al., 2019)" was corrected to "With high abundance (~$10^{17}$ cm$^{-3}$) being detected in the troposphere (Huang et al., 2015), $H_2O$ has been reported to effectively act as a catalyst in chemical reactions (Liu et al., 2019)." in line 171 page 6 of our revised manuscript.

*3. Line 369, 'sight'. I guess the authors mean 'slight'?*

Author reply:

Thanks for the referee's suggestion. According to the suggestion, we corrected the above spelling error. The sentence of "Given the fact that GA has only a sight influence on the nucleation and growth processes of atmospheric clusters, the reaction between GA and $SO_3$ may provide a secondary source of the potential precursor since high concentrations of sulfur oxides being detected." in line 398 page 16 was corrected to "Given the fact that GA has only a slight influence on the nucleation and growth processes of atmospheric clusters, the reaction between GA and $SO_3$ may provide a secondary source of the potential precursor since high concentrations of sulfur oxides being detected." in our revised manuscript.

We appreciate the reviewer's suggestions very much improving our presentation.

**Response to referee #2:**

Referee's Comments to Author:

*The paper investigates if organic acids, organic sulfates or organic sulfuric anhydrides could enhance new particle formation driven only by sulfuric acid (SA) and dimethyl amine (DMA). The authors present quantum chemical calculations of the reaction of $SO_3$ with glycolic acid to glycolic acid sulfate (GAS) and glycolic acid sulfuric anhydride (GASA). They demonstrate that the addition of a catalyst (e.g. water) makes this reaction almost barrierless and thus could be a potential pathway to form GAS and GASA in the gas phase. Furthermore, lowest free energy structures of (GA)x-(SA)y-(DMA)z, (GAS)x-(SA)y-(DMA)z, and (GASA)x-(SA)y-(DMA)z clusters, their formation Gibbs Free Energies and evaporation rates have been calculated. It is shown that (GA)x-(SA)y-(DMA)z clusters are least stable, while mixed clusters with GAS and GASA are similar to or more stable than pure SA-DMA clusters. In a next step the authors calculate mixed cluster formation rates and determine an enhancement effect of GAS on SA-DMA driven new particle formation (NPF). Cluster growth pathways are then shown for different concentrations of SA, GAS and DMA. Finally, the authors compare their theoretical results with ambient observations at Mount Tai in China and conclude, that GAS could explain deviations between pure SA-DMA driven NPF and observations. Furthermore, they propose that the formation of organosulfates by this gas phase reaction may be a source of organosulfates often observed in secondary organic aerosols.*

*The first part of the manuscript including the quantum chemical calculations of the thermodynamic parameters of the organosulfates, as well as the mixed cluster geometries and stabilities is fine. However, the second part has some serious issues. There are no quantitative measurements of GAS and GASA in the atmosphere. Thus, the authors assume for $SO_3$ a concentration of $10^5$ $cm^{-3}$ and a range of measured ambient GA concentrations to calculate potential equilibrium concentrations for GAS and GASA. GASA levels would be negligibly small while GAS concentrations could reach $10^3$-$10^5$ $cm^{-3}$. The authors do not give a reference for their assumed $SO_3$ concentration of $10^5$ $cm^{-3}$.*

Author reply:

Thanks for the referee's question and comments. In order to reconsider the issue of GAS and GASA concentrations in the atmosphere more carefully, we did our best to search for reports on their measurements again. Due to the current lack of observations of organic sulfates in gas phase in the atmosphere, we found only one relevant report. Ehn et al. have made the observation of an organosulfate (glycolic acid sulfate, GAS) in the gas phase, and reported its ion concentration in the gas phase (6.7 molecules $cm^{-3}$) in the boreal forest (Ehn et al., 2010). In the same observation, the gas phase $H_2SO_4$ (SA) ion concentration was identified to be 242.7 molecules $cm^{-3}$, which is two orders

of magnitude higher than that of GAS. Unfortunately, we did not find more reports on gas phase concentrations of neutral organosulfates. For the concentration of SA, Kulmala et al., Weber et al. and other researchers have reported that the typical tropospheric concentration of ambient sulfuric acid range from $10^5$ to $10^7$ molecules cm$^{-3}$ (Kulmala et al., 2000;Weber et al., 1999;Weber et al., 1998;Riipinen et al., 2007). In Ehn et al.'s observation (Ehn et al., 2010), they found the ion concentration of GAS is two orders of magnitude lower than that of SA in the gas phase. If it is assumed that the concentrations of their neutral species have similar proportions, the concentration of gas phase GAS in the atmosphere can be estimated to be ~ $10^3$ to $10^5$ molecules cm$^{-3}$, which agrees well with the GAS calculated concentrations in our work.

For $SO_3$ concentrations in the atmosphere, the ambient $SO_3$ was detected by Yao et al. from two field measurements in urban Beijing in the atmosphere, and showed that the concentration of $SO_3$ varied from ~$4.0 \times 10^4$ to $1.9 \times 10^6$ molecules cm$^{-3}$ during the winter (Yao et al., 2020). Many works, which investigated the reaction mechanisms of organic compounds and $SO_3$ at the aerosol surface by Tan et al., Zhong et al. and Liu et al., have theoretically evaluated the concentration of organic sulfate considering $SO_3$ concentration at $10^5$ molecules cm$^{-3}$ (Tan et al., 2022; Zhong et al., 2019; Liu et al., 2019). Tsona Tchinda et al. discussed the catalyzed effect of organic acid on the pyruvic acid-catalyzed $SO_3$ hydrolysis at the $SO_3$ concentration of $10^5$ molecules cm$^{-3}$ (Tsona Tchinda et al., 2022). Li et al. has also investigated the self-catalytic reaction of $SO_3$ and $NH_3$ in the atmosphere at the $SO_3$ concentration of $10^5$ molecules cm$^{-3}$ (Li et al., 2018).

Considering above measurements and theoretical publications about atmospheric $SO_3$, we think $10^5$ molecules cm$^{-3}$ is a reasonable concentration for $SO_3$ investigated in our work. According the referee's comments and concerns, the following sentences and references were modified and added in page 5 Line 121 in our revised manuscript.

"We use the reactant concentrations of [GA] = $1.11 \times 10^7$-$2.72 \times 10^9$ molecules cm$^{-3}$ according to the values of some field observations (Mochizuki et al., 2019; Miyazaki et al., 2014; Stieger et al., 2021; Mochizuki et al., 2017). Considering field measurements (Yao et al., 2020) and theoretical investigations (Tan et al., 2022; Zhong et al., 2019; Liu et al., 2019; Tsona Tchinda et al., 2022; Li et

al., 2018) of atmospheric $SO_3$, its concentration is assumed to be $10^5$ molecules cm$^{-3}$ here. Based on the above equations, the estimated concentration of the reaction product, GAS, is about $2.14 \times 10^3$-$5.24 \times 10^5$ molecules cm$^{-3}$, and GASA is about $2.30 \times 10^{-6}$-$5.62 \times 10^{-4}$ molecules cm$^{-3}$."

*In fact, $SO_3$ is very rapidly converted by water vapor to sulfuric acid, which in turn is condensing on aerosol. Let's assume a condensation sink for SA of 0.01 s$^{-1}$, $k(s^{-1})$ = 3.90 x 10$^{-41}$ exp(6830.6/T)[$H_2O$]$^2$ (J.Phys. Chem. A 1997, 101,10000-10011), [$H_2O$]= 2·10$^{17}$ cm$^{-3}$, an $SO_3$ level of 10$^5$ cm$^{-3}$ would then yield a steady state concentration of SA = 5.6 ppb. Ambient SA levels are at least a factor 1000 lower. Therefore, possible GAS concentrations would be much lower in the ambient. Furthermore, $SO_3$ and SA scale with each other. It is very unlikely that [$SO_3$]=[SA]=10$^5$cm$^{-3}$ as assumed in Figure 3. The simulated cluster formation rates in Figures 3 and S6 are only a theoretical exercise but not at all relevant for ambient conditions.*

Author reply:

Thanks for the referee's comments. Since the main source of sulfuric acid in the atmosphere is the reaction of $SO_3$ and $H_2O$, considering this reaction to evaluate the concentration of atmospheric $SO_3$ is a good point. As reported, the reaction of $SO_3$ with water vapor is believed to be the principal mechanism for gas phase sulfuric acid ($H_2SO_4$) formation in the atmosphere (Stockwell and Calvert, 1983;Castleman Jr et al., 1975;Kolb et al., 1994). The relevant reaction equation between $SO_3$ and water ($H_2O$) can be described by:

$$SO_3 + H_2O + M \rightleftharpoons SO_3 \cdots H_2O + M$$

$$SO_3 \cdots H_2O \rightarrow H_2SO_4$$

According to both of the theoretical calculations and laboratory experiments, the direct reaction between $SO_3$ and $H_2O$ is thermodynamically unfavorable owing to the high energy barrier, clearly indicating that a facilitator molecule M, acting as a catalyst, is required in the gas-phase reaction (Kolb et al., 1994;Lovejoy et al., 1996;Morokuma and Muguruma, 1994;Jayne et al., 1997;Torrent-Sucarrat et al., 2012;Hazra and Sinha, 2011;Bandyopadhyay et al., 2017). Theoretical calculations further revealed that this above-mentioned mechanism, involving a four-member ring transition state, has a large activation energy barrier ($\sim$28 to 32 kcal mol$^{-1}$) and consequently was not favored as a possible route for atmospheric $H_2SO_4$ production (Chen and Plummer, 1985;Hofmann and Schleyer, 1994;Morokuma and Muguruma, 1994;Steudel, 1995). Hazra et al. have demonstrated the presence of

formic acid could substantially reduce the energy barrier between $SO_3$ and $H_2O$ (Hazra and Sinha, 2011). We have carefully read the reference (J. Phys. Chem. A 1997, 101, 10000-10011) the referee mentioned. We are very sorry for not understanding the calculations. For the equation $k(s^{-1}) = 3.90 \times 10^{-41} \exp(6830.6/T)[H_2O]^2$, it is a experimentally obtained rate constant equation for the reaction between $SO_3$ and $H_2O$ to form $H_2SO_4$. We did not get how to obtain a steady state concentration of SA from $H_2O$ and $SO_3$ concentration according to this equation. Hence, we try to estimate the concentration of SA by the following method. Given above previous identified mechanisms, the formation of $H_2SO_4$ can be described by following reaction:

$$H_2O + SO_3 \rightarrow H_2SO_4$$

The equilibrium constant $K_{eq}$ for the formation of $H_2SO_4$ is

$$K_{eq} = \frac{[H_2SO_4]}{[H_2O][SO_3]} = e^{\frac{-\Delta G}{RT}}$$

And the equilibrium concentration of $H_2SO_4$ can be roughly estimated theoretically using the following expression:

$$[H_2SO_4] = K_{eq}[H_2O][SO_3]$$

where $K_{eq}$ is equal to the equilibrium constant from the formation Gibbs energy of the $H_2SO_4$. According to the results calculated by Liu et al., the Gibbs free energy barrier of reaction between $SO_3$ and $H_2O$ is 24.11 kcal mol$^{-1}$ at 280 K (Liu et al., 2019). $[H_2O]$ and $[SO_3]$ are the concentrations of $H_2O$ and $SO_3$ monomer, respectively. According to the reactant concentrations suggested by referee and previous work, the concentration of $[H_2O]$ is $2 \times 10^{17}$ molecules cm$^{-3}$, and that of $[SO_3]$ is $10^5$ molecules cm$^{-3}$ (~0.0037 ppt). Based on the above equations, the estimated concentration of the reaction product, $H_2SO_4$, is about $7 \times 10^4$ molecules cm$^{-3}$, which is quite close to the reported SA concentration in range from $10^5$ to $10^7$ molecules cm$^{-3}$ (Kulmala et al., 2000;Weber et al., 1999;Weber et al., 1998;Riipinen et al., 2007). This result also suggests $10^5$ molecules cm$^{-3}$ is reasonable for the $SO_3$ concentration in the atmosphere.

In Figure 3, it is not mean that $10^5$ molecules cm$^{-3}$ SA is generated from $10^5$ molecules cm$^{-3}$ $SO_3$. In the atmosphere, species like SA, GA, $SO_3$, and so on all should be in a certain concentration range. Since we are trying to discuss the nucleation of the ternary system, we can only assume that two of the

species are reasonably certain values, and discuss the concentration of the third species within a certain range. Hence, the condition of [DMA] = $10^8$ molecules cm$^{-3}$ and [SA] = $10^5$ molecules cm$^{-3}$ were assumed for the discussion. To discuss the concentration of GAS in a range, we use the reactant concentrations of [GA] = $1.11 \times 10^7$-$2.72 \times 10^9$ molecules cm$^{-3}$ according to the values of some field observations (Mochizuki et al., 2019; Miyazaki et al., 2014; Stieger et al., 2021; Mochizuki et al., 2017). The concentration of SO$_3$ is assumed to be $10^5$ molecules cm$^{-3}$ here. The estimated concentration of the reaction product, GAS, is about $2.14 \times 10^3$-$5.24 \times 10^5$ molecules cm$^{-3}$ as displayed in Figure 3. The results for different SA concentrations were also given in Figure S8 in our revised manuscript.

[Figure]

Figure S8. Simulated cluster formation rates $J$ (cm$^{-3}$s$^{-1}$) as a function of monomer concentrations ([GA], [GAS], and [GASA], respectively) under different [SA] (a) (d) [SA] = $10^4$, (b) (e) [SA] = $10^6$, and (c) (f) [SA] = $10^7$ molecules cm$^{-3}$ at 278 K, [DMA] = $10^8$ molecules cm$^{-3}$.

In order to further consider the effect of different concentrations of SO$_3$ according to the referee's comments, we investigated the formation rates ($J_{GAS-SA-DMA}$ and $J_{GA-SA-DMA}$) at the conditions of varying level [SO$_3$] ([SO$_3$] = $10^4$ molecules cm$^{-3}$, [SO$_3$] = $10^5$ molecules cm$^{-3}$, and [SO$_3$] = $10^6$ molecules cm$^{-3}$). The following results and descriptions were added in page S14 in the revised SI of

our manuscript. "As the results displayed in Figure S9, we found that the cluster formation rate of GAS-SA-DMA system markedly increases with the increasing concentration of [GAS] compared to that of GA-SA-DMA system, especially in the case of [$SO_3$] = $10^6$ molecules cm$^{-3}$, which $J_{GAS-SA-DMA}$ could be up to 2 orders of magnitude higher then $J_{GA-SA-DMA}$."

[Figure]

**Figure S9**. Simulated cluster formation rates $J$ (cm$^{-3}$s$^{-1}$) as a function of monomer concentrations ([GA] and [GAS]) at 278 K and [$SO_3$] = $10^4$ molecules cm$^{-3}$ (left panel) [$SO_3$] = $10^5$ molecules cm$^{-3}$ (center panel) [$SO_3$] = $10^6$ molecules cm$^{-3}$ (right panel) under the condition of [DMA] = $10^8$ molecules cm$^{-3}$ and [SA] = $10^5$ molecules cm$^{-3}$.

*The authors also argue that measured new particle formation rates at Mt. Tai could not be explained by pure SA-DMA nucleation. For the comparison the authors use theoretical NPF-rates from ACDC calculations at a cluster size of only about 1.2-1.4 nm, while the measurements were made at 3 nm. The authors apparently assume that the nucleation rate at the two different cluster sizes should be the same. That is not at all the case. J(3nm) is probably more than a factor of 10 slower than J(1.3nm) (see e.g. Xiao et al., ACP 21, 14275–14291, 2021). Thus, this comparison does not provide evidence that GAS could explain the fast NPF rate.*

Author reply:

Thanks for the referee's comments on the consideration of the different nucleation rate for different cluster size. Due to the lack of accurate observational data, the relationship between the formation rates of particles of different sizes under different environmental conditions is difficult to accurately quantify. In Xiao et al.'s work (ACP 21, 14275–14291, 2021), they calculated the ratio between the formation rates at 2.5 and 1.7 nm as the survival probability in CLOUD chamber experimmets. In their experiments, it is obvious that the survival probability is closely related to various experimental conditions such as nucleation precursor species and concentrations, temperature, condensation sink and so on. Unfortunately they did not give a quantitative relationship between the formation rates of clusters of different sizes, so we cannot use it to correct our calculated results. In order to consider this

issue more rigorously, we searched the relevant literatures. According to the revised Kerminen−Kulmala equation (Anttila et al., 2010;Lehtinen et al., 2007;Kulmala et al., 2012), cluster formation rates for 3.0 nm clusters ($J_{3.0}$) relate to those for 1.2-1.4 nm clusters ($J_{1.2}$) by

$$J_{1.2} = J_{3.0}\exp[\gamma\left(\frac{1}{1.2} - \frac{1}{3.0}\right)\frac{CS}{GR}]$$

where GR is the initial cluster growth rate from 1.0 to 3.0 nm, CS represents condensation sink of clusters by preexisting particles and $\gamma$ is a coefficient with a value of approximately 0.23 $m^2\,nm^2\,h^{-1}$ (Riipinen et al., 2007;Xia et al., 2020). GR was measured to be 1.5−3.1 $nm\cdot h^{-1}$ in the 1.0−3.0 nm size range during the observed events (Riipinen et al., 2007). CS was between 0.01 and 0.04 $s^{-1}$ (Xia et al., 2020). $J_{1.2}$ was then calculated to be 1.0003−1.0031 times of $J_{3.0}$. This result suggests that the $J_{1.2}$ and $J_{3.0}$ are close at above conditions, which are similar with that considered in our work. However, because always $J_{1.2} > J_{3.0}$, using the $J_{3.0}$ value instead of $J_{1.2}$ should lead to slight underestimation of $J_{1.2}$. As precursor concentrations ([$H_2SO_4$], [DMA], and [GAS]) positively correlate with $J_{GAS-SA-DMA}$, $J_{GA-SA-DMA}$ and/or $J_{GA-SA-DMA}$, the required precursor concentrations calculated by ACDC simulations for the observed 3.0 nm clusters may be slightly underestimated. The following sentences were added in our revised manuscript and SI respectively.

"Noted that the calculated cluster formation rates via ACDC in this work are at a cluster size about 1.2-1.4 nm. The observed cluster formation rates ($J_{obs}$) at Mt. Tai were measured at 3 nm. According to the revised Kerminen−Kulmala equation (Anttila et al., 2010;Lehtinen et al., 2007;Kulmala et al., 2012), the cluster formation rate for 1.2 nm cluster ($J_{1.2}$) is slightly larger than that for 3.0 nm clusters ($J_{3.0}$) ($J_{1.2}$ is calculated to be 1.0003−1.0031 times of $J_{3.0}$). The above required precursor concentrations calculated by ACDC simulations for the observed 3.0 nm clusters may be slightly underestimated (see details in the SI)." were added in page 18 line 436 in our modified manuscript.

"According to the revised Kerminen−Kulmala equation (Anttila et al., 2010;Lehtinen et al., 2007;Kulmala et al., 2012), cluster formation rates for 3.0 nm clusters ($J_{3.0}$) relate to those for 1.2-1.4 nm clusters ($J_{1.2}$) by

$$J_{1.2} = J_{3.0}\exp[\gamma\left(\frac{1}{1.2} - \frac{1}{3.0}\right)\frac{CS}{GR}]$$

where GR is the initial cluster growth rate from 1.0 to 3.0 nm, CS represents condensation sink of clusters by preexisting particles and $\gamma$ is a coefficient with a value of approximately 0.23 $m^2\,nm^2\,h^{-1}$ (Riipinen et al., 2007;Xia et al., 2020). GR was measured to be 1.5−3.1 nm·h$^{-1}$ in the 1.0−3.0 nm size range during the observed events (Riipinen et al., 2007). CS was between 0.01 and 0.04 s$^{-1}$ (Xia et al., 2020). $J_{1.2}$ was then calculated to be 1.0003−1.0031 times of $J_{3.0}$. This result suggests that the $J_{1.2}$ and $J_{3.0}$ are close at above conditions, which are similar with that considered in our work. However, because always $J_{1.2} > J_{3.0}$, using the $J_{3.0}$ value instead of $J_{1.2}$ should lead to slight underestimation of $J_{1.2}$. As precursor concentrations ([H$_2$SO$_4$], [DMA], and [GAS]) positively correlate with $J_{GAS-SA-DMA}$, $J_{GA-SA-DMA}$ and/or $J_{GA-SA-DMA}$, the required precursor concentrations calculated by ACDC simulations for the observed 3.0 nm clusters may be slightly underestimated." were added in page S4 line 61 in our modified SI.

*Overall, I think the second part of the paper about the potential role and importance of GAS for ambient NPF is untenable. I do not see how this hypothesis could be substantiated. Since the first part is only QC calculations, I think the first part alone is not suited for ACP.*
*Minor comments*
*Line 57: it is not proven so far that SO$_3$ is a major oxidant in the atmosphere. It is also not emitted but formed as an intermediate species through oxidation of SO$_2$.*

Author reply:

Thanks for the referee's comments. For SO$_3$ concentration in the atmosphere, the ambient SO$_3$ was detected by Yao et al. from two field measurements in urban Beijing in the atmosphere, and showed that the concentration of SO$_3$ varied from ~4.0 × 10$^4$ to 1.9 × 10$^6$ molecules cm$^{-3}$ during the winter (Yao et al., 2020). It has been reported that sulfur trioxide (SO$_3$) is a major air pollutant and is mainly produced by the gas-phase oxidation of SO$_2$ (Stockwell and Calvert, 1983;Mauldin Iii et al., 2012;Zhong et al., 2017;Li et al., 2018). Furthermore, SO$_3$ is a highly reactive gas and one of the most common acid oxides (Fleig et al., 2012), which can lead to both acid rain and atmospheric aerosol (Sipila et al., 2010;England et al., 2000).

*Figure 3: For J rates you should say at what cluster size they have been calculated*

Author reply:

Thanks for the referee's suggestions. In order to clarify this issue, following sentences were added in page 18 line 436 in our modified manuscript. "Noted that the calculated cluster formation rates via ACDC in this work are at a cluster size about 1.2-1.4 nm."

*Line 311: high mountain sites and polar regions are usually not places with high amine concentrations, do you have references?*

Author reply:

We appreciated the referee's comments and questions. Liu et al. observed high amines concentrations in gas phase which could be up to $307\pm196.7$ ng m$^{-3}$ at Nanling Mountains, southern China in summer (298 K) (Liu et al., 2018). Specifically, the measured dimethylamine concentration is up to 300 ng m$^{-3}$ ($\sim4 \times 10^9$ molecules cm$^{-3}$). And a forested site on the northern foot of Mt. Fuji, Japan has measured the dimethylamine with the concentration of 226.5 pmol m$^{-3}$ (Matsumoto et al., 2023). Baumgardner et al. have reported high atmospheric amine concentrations and amines were quantified with concentrations of 0.36-1.42 μg m$^{-3}$ from the mountain site, and other research groups have also measured high amine concentration in the atmosphere and/or particulate matter (Ge et al., 2011;Kürten et al., 2014;Drewnick et al., 2007;Baumgardner et al., 2009;Roth et al., 2016;Liu et al., 2023). Of particular note, alkylamines in the surface ocean and atmosphere of the Antarctic sympagic environment have been detected from 0.01 ng m$^{-3}$ to 7.1 ng m$^{-3}$ (Dall'Osto et al., 2019). Baumgardner et al. have reported high atmospheric amine concentrations and amines were quantified with concentrations of 0.36-1.42 μg m$^{-3}$ from the mountain site (Ge et al., 2011;Kürten et al., 2014;Drewnick et al., 2007;Baumgardner et al., 2009). Following sentences and citations were modified and added in page 10 line 264 in our modified manuscript. "The concentration of DMA is selected to be $10^8$ molecules cm$^{-3}$, according to the typical concentrations observed in the gas phase at high mountains (Liu et al., 2018b, Matsumoto et al., 2023)."

*Figure S1: there is no red line. The blue line should read green. "the pathway to form H₂SO₄ with as a catalyst"   should read "the pathway to form H₂SO₄ with H₂O as a catalyst"*

Author reply:

Thanks for the referee's suggestion. According to the suggestion, we corrected the above spelling error. The caption of Figure S1 "The red line represents the pathway through $SO_3$ attacking the -OH group of GA with $H_2O$ as a catalyst;" was corrected to "The green line represents the pathway through $SO_3$ attacking the -OH group of GA with $H_2O$ as a catalyst;" in our revised supporting information for the manuscript.

*Table S1: how do you get the ΔG values from Figure 1?*

Author reply:

Thanks for the referee's question. To clarify the data of our calculation results, the ΔG values under different temperatures (278 K, Table S4 and 298 K, Table S5) were added in page S20 and S21 in our revised SI of manuscript as below.

**Table S4**. Calculated Gibbs free energy changes ($\Delta G$) of the formation of heterotrimers consisting of $H_2SO_4$, base (ammonia/DMA), and GA/GAS/GASA at the temperature of 278 K and pressure of 101.3 KPa.

| clusters | $\Delta G$ (kcal mol$^{-1}$) GA-SA-ammonia [a] | $\Delta G$ (kcal mol$^{-1}$) GA-SA-DMA | $\Delta G$ (kcal mol$^{-1}$) GAS-SA-DMA | $\Delta G$ (kcal mol$^{-1}$) GASA-SA-DMA |
|---|---|---|---|---|
| Org-base | -2.74 | -4.23 | -7.83 | -3.06 |
| Org-SA | -7.55 | -7.97 | -2.58 | -5.27 |
| Org-SA-base | -14.90 | -23.12 | -29.90 | -34.54 |
| Org-SA-2base | -16.68 | -32.66 | -50.48 | -52.57 |
| Org-2SA | -14.55 | -11.70 | -13.58 | -15.75 |
| Org-2SA-base | -28.21 | -37.71 | -41.62 | -40.26 |
| Org-2SA-2base | -36.81 | -55.95 | -65.39 | -71.31 |
| Org-2SA-3base | -41.21 | -67.68 | -82.56 | -90.30 |
| 2Org | -5.37 | -5.17 | -6.68 | -0.77 |
| 2Org-base | -4.33 | -10.05 | -18.20 | -27.65 |
| 2Org-2base | -2.35 | -11.59 | -50.36 | -64.19 |

| clusters | | | | |
|---|---|---|---|---|
| 2Org-SA | -14.87 | -15.04 | -16.86 | -14.54 |
| 2Org-SA-base | -19.96 | -26.48 | -38.94 | -44.23 |
| 2Org-SA-2base | -20.99 | -39.72 | -62.20 | -68.50 |
| 2Org-SA-3base | -23.42 | -44.48 | -76.48 | -90.80 |
| 3Org | -5.24 | -1.81 | -7.71 | -11.82 |
| 3Org-base | -6.87 | -13.61 | -29.49 | -40.64 |
| 3Org-2base | -6.98 | -20.11 | -54.73 | -69.36 |
| 3Org-3base | -3.20 | -27.43 | -78.18 | -98.00 |

**Table S5**. Calculated Gibbs free energy changes ($\Delta G$) of the formation of heterotrimers consisting of $H_2SO_4$, base (ammonia/DMA), and GA/GAS/GASA at the temperature of 298 K and pressure of 101.3 KPa.

| clusters | $\Delta G$ (kcal mol$^{-1}$) SA-GA-ammonia [a] | $\Delta G$ (kcal mol$^{-1}$) SA-GA-DMA | $\Delta G$ (kcal mol$^{-1}$) SA-GAS-DMA | $\Delta G$ (kcal mol$^{-1}$) SA-GASA-DMA |
|---|---|---|---|---|
| Org-base | -2.11 | -3.50 | -7.10 | -2.37 |
| Org-SA | -6.83 | -7.28 | -1.78 | -4.51 |
| Org-SA-base | -13.51 | -21.76 | -28.35 | -32.90 |
| Org-SA-2base | -14.55 | -30.36 | -48.13 | -50.06 |
| Org-2SA | -12.94 | -10.10 | -11.86 | -14.04 |
| Org-2SA-base | -25.95 | -35.22 | -39.15 | -37.64 |
| Org-2SA-2base | -33.76 | -52.74 | -62.14 | -67.93 |
| Org-2SA-3base | -37.48 | -63.71 | -78.55 | -86.19 |
| 2Org | -4.62 | -4.40 | -5.76 | 0.13 |
| 2Org-base | -2.97 | -8.64 | -16.54 | -25.84 |
| 2Org-2base | -0.32 | -9.03 | -47.82 | -61.62 |
| 2Org-SA | -13.39 | -13.60 | -15.13 | -12.75 |
| 2Org-SA-base | -17.74 | -24.08 | -36.49 | -41.54 |
| 2Org-SA-2base | -17.78 | -36.51 | -58.92 | -65.15 |
| 2Org-SA-3base | -19.65 | -40.59 | -72.36 | -86.50 |
| 3Org | -3.76 | -0.12 | -5.99 | -10.01 |

| | | | | |
|---|---|---|---|---|
| 3Org-base | -4.67 | -11.16 | -26.82 | -37.95 |
| 3Org-2base | -3.94 | -16.80 | -51.34 | -65.78 |
| 3Org-3base | -17.78 | -23.60 | -74.01 | -93.69 |

We appreciate the referee's comments and suggestions for clarifying and improving our presentation.

**Reference**

Anttila, T., Kerminen, V.-M., and Lehtinen, K. E. J.: Parameterizing the formation rate of new particles: The effect of nuclei self-coagulation, J. Aerosol Sci., 41, 621-636, 10.1016/j.jaerosci.2010.04.008, 2010.

Bandyopadhyay, B., Kumar, P., and Biswas, P.: Ammonia Catalyzed Formation of Sulfuric Acid in Troposphere: The Curious Case of a Base Promoting Acid Rain, J. Phys. Chem. A, 121, 3101-3108, 10.1021/acs.jpca.7b01172, 2017.

Baumgardner, D., Grutter, M., Allan, J., Ochoa, C., Rappenglueck, B., Russell, L. M., and Arnott, P.: Physical and chemical properties of the regional mixed layer of Mexico's Megapolis, Atmos. Chem. Phys., 9, 5711-5727, 10.5194/acp-9-5711-2009, 2009.

Castleman Jr, A., Davis, R. E., Munkelwitz, H., Tang, I., and Wood, W. P.: Kinetics of association reactions pertaining to $H_2SO_4$ aerosol formation, Int. J. Chem. Kinet.;(United States), 7, 1975.

Chen, T., and Plummer, P. L.: Ab initio MO investigation of the gas-phase reaction sulfur trioxide+ water → sulfuric acid, J. Phys. Chem., 89, 3689-3693, 1985.

Dall'Osto, M., Airs, R. L., Beale, R., Cree, C., Fitzsimons, M. F., Beddows, D., Harrison, R. M., Ceburnis, D., O'Dowd, C., Rinaldi, M., Paglione, M., Nenes, A., Decesari, S., and Simó, R.: Simultaneous Detection of Alkylamines in the Surface Ocean and Atmosphere of the Antarctic Sympagic Environment, ACS Earth Space Chem., 3, 854-862, 10.1021/acsearthspacechem.9b00028, 2019.

Drewnick, F., Schneider, J., Hings, S. S., Hock, N., Noone, K., Targino, A., Weimer, S., and Borrmann, S.: Measurement of Ambient, Interstitial, and Residual Aerosol Particles on a Mountaintop Site in Central Sweden using an Aerosol Mass Spectrometer and a CVI, J. Atmos. Chem., 56, 1-20, 10.1007/s10874-006-9036-8, 2007.

Ehn, M., Junninen, H., Petäjä, T., Kurtén, T., Kerminen, V. M., Schobesberger, S., Manninen, H. E., Ortega, I. K., Vehkamäki, H., Kulmala, M., and Worsnop, D. R.: Composition and temporal behavior of ambient ions in the boreal forest, Atmos. Chem. Phys., 10, 8513-8530, 10.5194/acp-10-8513-2010, 2010.

England, G. C., Zielinska, B., Loos, K., Crane, I., and Ritter, K.: Characterizing PM2.5 emission profiles for stationary sources: comparison of traditional and dilution sampling techniques, Fuel Process. Technol., 65-66, 177-188, 10.1016/S0378-3820(99)00085-5, 2000.

Fleig, D., Vainio, E., Andersson, K., Brink, A., Johnsson, F., and Hupa, M.: Evaluation of $SO_3$ Measurement Techniques in Air and Oxy-Fuel Combustion, Energy & Fuels, 26, 5537-5549, 10.1021/ef301127x, 2012.

Ge, X. L., Wexler, A. S., and Clegg, S. L.: Atmospheric amines - Part I. A review, Atmos. Environ., 45, 524-546, 10.1016/j.atmosenv.2010.10.012, 2011.

Hazra, M. K., and Sinha, A.: Formic Acid Catalyzed Hydrolysis of $SO_3$ in the Gas Phase: A Barrierless Mechanism for Sulfuric Acid Production of Potential Atmospheric Importance, J. Am. Chem. Soc., 133, 17444-17453, 10.1021/ja207393v, 2011.

Hofmann, M., and Schleyer, P. v. R.: Acid rain: Ab initio investigation of the $H_2O$. cntdot. $SO_3$ complex and its conversion to $H_2SO_4$, J. Am. Chem. Soc., 116, 4947-4952, 1994.

Huff, A. K., Mackenzie, R. B., Smith, C. J., and Leopold, K. R.: Facile Formation of Acetic Sulfuric Anhydride: Microwave Spectrum, Internal Rotation, and Theoretical Calculations, J. Phys. Chem. A, 121, 5659-5664, 10.1021/acs.jpca.7b05105, 2017.

Huff, A. K., Mackenzie, R. B., Smith, C. J., and Leopold, K. R.: A Perfluorinated Carboxylic Sulfuric Anhydride: Microwave and Computational Studies of $CF_3COOSO_2OH$, J. Phys. Chem. A, 123, 2237-2243, 10.1021/acs.jpca.9b00300, 2019.

Jayne, J. T., Pöschl, U., Chen, Y.-m., Dai, D., Molina, L. T., Worsnop, D. R., Kolb, C. E., and Molina, M. J.: Pressure and Temperature Dependence of the Gas-Phase Reaction of $SO_3$ with $H_2O$ and the Heterogeneous Reaction of $SO_3$ with $H_2O/H_2SO_4$ Surfaces, J. Phys. Chem. A, 101, 10000-10011, 10.1021/jp972549z, 1997.

Kolb, C., Jayne, J., Worsnop, D., Molina, M., Meads, R., and Viggiano, A.: Gas phase reaction of sulfur trioxide with water vapor, J. Am. Chem. Soc., 116, 10314-10315, 1994.

Kulmala, M., Pirjola, L., and Mäkelä, J. M.: Stable sulphate clusters as a source of new atmospheric particles, Nature, 404, 66-69, 2000.

Kulmala, M., Petäjä, T., Nieminen, T., Sipilä, M., Manninen, H. E., Lehtipalo, K., Dal Maso, M., Aalto, P. P., Junninen, H., Paasonen, P., Riipinen, I., Lehtinen, K. E. J., Laaksonen, A., and Kerminen, V.-M.: Measurement of the nucleation of atmospheric aerosol particles, Nat. Protoc., 7, 1651-1667, 10.1038/nprot.2012.091, 2012.

Kürten, A., Jokinen, T., Simon, M., Sipilä, M., Sarnela, N., Junninen, H., Adamov, A., Almeida, J., Amorim, A., Bianchi, F., Breitenlechner, M., Dommen, J., Donahue, N. M., Duplissy, J., Ehrhart, S., Flagan, R. C., Franchin, A., Hakala, J., Hansel, A., Heinritzi, M., Hutterli, M., Kangasluoma, J., Kirkby, J., Laaksonen, A., Lehtipalo, K., Leiminger, M., Makhmutov, V., Mathot, S., Onnela, A., Petäjä, T., Praplan, A. P., Riccobono, F., Rissanen, M. P., Rondo, L., Schobesberger, S., Seinfeld, J. H., Steiner, G., Tomé, A., Tröstl, J., Winkler, P. M., Williamson, C., Wimmer, D., Ye, P., Baltensperger, U., Carslaw, K. S., Kulmala, M., Worsnop, D. R., and Curtius, J.: Neutral molecular cluster formation of sulfuric acid–dimethylamine observed in real time under atmospheric conditions, Proc. Natl. Acad. Sci., 111, 15019-15024, 10.1073/pnas.1404853111, 2014.

Lehtinen, K. E. J., Dal Maso, M., Kulmala, M., and Kerminen, V.-M.: Estimating nucleation rates from apparent particle formation rates and vice versa: Revised formulation of the Kerminen–Kulmala equation, J. Aerosol Sci., 38, 988-994, 10.1016/j.jaerosci.2007.06.009, 2007.

Li, H., Kupiainen-Määttä, O., Zhang, H., Zhang, X., and Ge, M.: A molecular-scale study on the role of lactic acid in new particle formation: Influence of relative humidity and temperature, Atmos. Environ., 166, 479-487, 10.1016/j.atmosenv.2017.07.039, 2017.

Li, H., Zhong, J., Vehkamäki, H., Kurtén, T., Wang, W., Ge, M., Zhang, S., Li, Z., Zhang, X., Francisco, J. S., and Zeng, X.: Self-catalytic reaction of $SO_3$ and $NH_3$ to produce sulfamic acid and its implication to atmospheric particle formation J. Am. Chem. Soc., 9, 10.1021/jacs.8b04928, 2018.

Liu, F., Bi, X., Zhang, G., Lian, X., Fu, Y., Yang, Y., Lin, Q., Jiang, F., Wang, X., Peng, P. a., and Sheng, G.: Gas-to-particle partitioning of atmospheric amines observed at a mountain site in southern China, Atmos. Environ., 195, 1-11, 10.1016/j.atmosenv.2018.09.038, 2018.

Liu, L., Zhong, J., Vehkamäki, H., Kurtén, T., Du, L., Zhang, X., Francisco, J. S., and Zeng, X.-C.: Unexpected quenching effect on new particle formation from the atmospheric reaction of methanol with $SO_3$, Proc. Natl. Acad. Sci. U.S.A., 6, 10.1073/pnas.1915459116, 2019.

Liu, L., Yu, F., Du, L., Yang, Z., Francisco, J. S., and Zhang, X.: Rapid sulfuric acid–dimethylamine nucleation enhanced by nitric acid in polluted regions, Proc. Natl. Acad. Sci., 118, e2108384118, 10.1073/pnas.2108384118, 2021.

Liu, T., Xu, Y., Sun, Q.-B., Xiao, H.-W., Zhu, R.-G., Li, C.-X., Li, Z.-Y., Zhang, K.-Q., Sun, C.-X., and Xiao, H.-Y.: Characteristics, Origins, and Atmospheric Processes of Amines in Fine Aerosol Particles in Winter in China, J. Geophys. Res.: Atmos., 128, e2023JD038974, 10.1029/2023JD038974, 2023.

Lovejoy, E. R., Hanson, D. R., and Huey, L. G.: Kinetics and Products of the Gas-Phase Reaction of $SO_3$ with Water, Journal of Phys. Chem., 100, 19911-19916, 10.1021/jp962414d, 1996.

Mackenzie, R. B., Dewberry, C. T., and Leopold, K. R.: Gas phase observation and microwave spectroscopic characterization of formic sulfuric anhydride, Science, 349, 58-61, 10.1126/science.aaa9704, 2015.

Matsumoto, K., Kuwabara, T., and Nakano, T.: Seasonal trends and potential sources of aliphatic amines in the aerosols and gas phase at a forested site on the northern foot of Mt. Fuji, Japan, Atmos. Environ., 309, 119885, 10.1016/j.atmosenv.2023.119885, 2023.

Mauldin Iii, R., Berndt, T., Sipilä, M., Paasonen, P., Petäjä, T., Kim, S., Kurtén, T., Stratmann, F., Kerminen, V.-M., and Kulmala, M.: A new atmospherically relevant oxidant of sulphur dioxide, Nature, 488, 193-196, 2012.

Morokuma, K., and Muguruma, C.: Ab initio molecular orbital study of the mechanism of the gas phase reaction $SO_3$+ $H_2O$: Importance of the second water molecule, J. Am. Chem. Soc., 116, 10316-10317, 1994.

Ning, A., Zhang, H., Zhang, X., Li, Z., Zhang, Y., Xu, Y., and Ge, M.: A molecular-scale study on the role of methanesulfinic acid in marine new particle formation, Atmos. Environ., 10.1016/j.atmosenv.2020.117378, 2020.

Olenius, T., Halonen, R., Kurtén, T., Henschel, H., Kupiainen-Määttä, O., Ortega, I. K., Jen, C. N., Vehkamäki, H., and Riipinen, I.: New particle formation from sulfuric acid and amines: Comparison of monomethylamine, dimethylamine, and trimethylamine, J. Geophys. Res.: Atoms., 122, 7103-7118, 10.1002/2017JD026501, 2017.

Riipinen, I., Sihto, S. L., Kulmala, M., Arnold, F., Dal Maso, M., Birmili, W., Saarnio, K., Teinilä, K., Kerminen, V. M., Laaksonen, A., and Lehtinen, K. E. J.: Connections between atmospheric sulphuric acid and new particle formation during QUEST III–IV campaigns in Heidelberg and Hyytiälä, Atmos. Chem. Phys., 7, 1899-1914, 10.5194/acp-7-1899-2007, 2007.

Rong, H., Liu, L., Liu, J., and Zhang, X.: Glyoxylic Sulfuric Anhydride from the Gas-Phase Reaction between Glyoxylic Acid and $SO_3$: A Potential Nucleation Precursor. J. Phys. Chem. A, 8, 10.1021/acs.jpca.0c01558, 2020.

Roth, A., Schneider, J., Klimach, T., Mertes, S., van Pinxteren, D., Herrmann, H., and Borrmann, S.: Aerosol properties, source identification, and cloud processing in orographic clouds measured by single particle mass spectrometry on a central European mountain site during HCCT-2010, Atmos. Chem. Phys., 16, 505-524, 10.5194/acp-16-505-2016, 2016.

Shen, J., Xie, H.-B., Elm, J., Ma, F., Chen, J., and Vehkamäki, H.: Methanesulfonic Acid-driven New Particle Formation Enhanced by Monoethanolamine: A Computational Study, Environ. Sci. Technol., 53, 14387-14397, 10.1021/acs.est.9b05306, 2019.

Shen, J., Elm, J., Xie, H.-B., Chen, J., Niu, J., and Vehkamäki, H.: Structural Effects of Amines in Enhancing Methanesulfonic Acid-Driven New Particle Formation, Environ. Sci. Technol., 54, 13498-13508, 10.1021/acs.est.0c05358, 2020.

Sipila, M., Berndt, T., Petaja, T., Brus, D., Vanhanen, J., Stratmann, F., Patokoski, J., Mauldin, R. L., Hyvarinen, A. P., Lihavainen, H., and Kulmala, M.: The Role of Sulfuric Acid in Atmospheric Nucleation, Science, 327, 1243-1246, 10.1126/science.1180315, 2010.

Smith, C., Huff, A. K., Ward, R. M., and Leopold, K. R.: Carboxylic Sulfuric Anhydrides, J. Phys. Chem. A, 124, 601-612, 2019.

Smith, C. J., Huff, A. K., Mackenzie, R. B., and Leopold, K. R.: Observation of Two Conformers of Acrylic Sulfuric Anhydride by Microwave Spectroscopy, J. Phys. Chem. A, 121, 9074-9080, 10.1021/acs.jpca.7b09833, 2017.

Steudel, R.: Sulfuric acid from sulfur trioxide and water—a surprisingly complex reaction, Angewandte Chemie International Edition in English, 34, 1313-1315, 1995.

Stockwell, W. R., and Calvert, J. G.: The mechanism of the HO-SO2 reaction, Atmos. Environ. (1967), 17, 2231-2235, 1983.

Tan, S., Zhang, X., Lian, Y., Chen, X., Yin, S., Du, L., and Ge, M.: OH Group Orientation Leads to Organosulfate Formation at the Liquid Aerosol Surface, J. Am. Chem. Soc., 10.1021/jacs.2c05807, 2022.

Torrent-Sucarrat, M., Francisco, J. S., and Anglada, J. M.: Sulfuric Acid as Autocatalyst in the Formation of Sulfuric Acid, J. Am. Chem. Soc., 134, 20632-20644, 10.1021/ja307523b, 2012.

Tsona, N. T., and Du, L.: Hydration of glycolic acid sulfate and lactic acid sulfate: Atmospheric implications, Atmos. Environ., 216, 116921, 10.1016/j.atmosenv.2019.116921, 2019.

Tsona Tchinda, N., Du, L., Liu, L., and Zhang, X.: Pyruvic acid, an efficient catalyst in $SO_3$ hydrolysis and effective clustering agent in sulfuric-acid-based new particle formation, Atmos. Chem. Phys., 22, 1951-1963, 10.5194/acp-22-1951-2022, 2022.

Weber, R. J., McMurry, P. H., Mauldin, L., Tanner, D. J., Eisele, F. L., Brechtel, F. J., Kreidenweis, S. M., Kok, G. L., Schillawski, R. D., and Baumgardner, D.: A study of new particle formation and growth involving biogenic and trace gas species measured during ACE 1, J. Geophys. Res.: Atoms., 103, 16385-16396, 10.1029/97JD02465, 1998.

Weber, R. J., McMurry, P. H., Mauldin III, R. L., Tanner, D. J., Eisele, F. L., Clarke, A. D., and Kapustin, V. N.: New Particle Formation in the Remote Troposphere: A Comparison of Observations at Various Sites, Geophys. Res. Lett., 26,

307-310, 10.1029/1998GL900308, 1999.

Xia, D., Chen, J., Yu, H., Xie, H.-b., Wang, Y., Wang, Z., Xu, T., and Allen, D. T.: Formation Mechanisms of Iodine–Ammonia Clusters in Polluted Coastal Areas Unveiled by Thermodynamics and Kinetic Simulations, Environ. Sci. Technol., 54, 9235-9242, 10.1021/acs.est.9b07476, 2020.

Xiao, M., Hoyle, C. R., Dada, L., Stolzenburg, D., Kürten, A., Wang, M., Lamkaddam, H., Garmash, O., Mentler, B., Molteni, U., Baccarini, A., Simon, M., He, X. C., Lehtipalo, K., Ahonen, L. R., Baalbaki, R., Bauer, P. S., Beck, L., Bell, D., Bianchi, F., Brilke, S., Chen, D., Chiu, R., Dias, A., Duplissy, J., Finkenzeller, H., Gordon, H., Hofbauer, V., Kim, C., Koenig, T. K., Lampilahti, J., Lee, C. P., Li, Z., Mai, H., Makhmutov, V., Manninen, H. E., Marten, R., Mathot, S., Mauldin, R. L., Nie, W., Onnela, A., Partoll, E., Petäjä, T., Pfeifer, J., Pospisilova, V., Quéléver, L. L. J., Rissanen, M., Schobesberger, S., Schuchmann, S., Stozhkov, Y., Tauber, C., Tham, Y. J., Tomé, A., Vazquez-Pufleau, M., Wagner, A. C., Wagner, R., Wang, Y., Weitz, L., Wimmer, D., Wu, Y., Yan, C., Ye, P., Ye, Q., Zha, Q., Zhou, X., Amorim, A., Carslaw, K., Curtius, J., Hansel, A., Volkamer, R., Winkler, P. M., Flagan, R. C., Kulmala, M., Worsnop, D. R., Kirkby, J., Donahue, N. M., Baltensperger, U., El Haddad, I., and Dommen, J.: The driving factors of new particle formation and growth in the polluted boundary layer, Atmos. Chem. Phys., 21, 14275-14291, 10.5194/acp-21-14275-2021, 2021.

Xu, C. X., Jiang, S., Liu, Y. R., Feng, Y. J., Wang, Z. H., Huang, T., Zhao, Y., Li, J., and Huang, W.: Formation of atmospheric molecular clusters of methanesulfonic acid-Diethylamine complex and its atmospheric significance, Atmos. Environ., 226, 9, 10.1016/j.atmosenv.2020.117404, 2020.

Yao, L., Fan, X., Yan, C., Kurtén, T., Daellenbach, K. R., Li, C., Wang, Y., Guo, Y., Dada, L., Rissanen, M. P., Cai, J., Tham, Y. J., Zha, Q., Zhang, S., Du, W., Yu, M., Zheng, F., Zhou, Y., Kontkanen, J., Chan, T., Shen, J., Kujansuu, J. T., Kangasluoma, J., Jiang, J., Wang, L., Worsnop, D. R., Petäjä, T., Kerminen, V.-M., Liu, Y., Chu, B., He, H., Kulmala, M., and Bianchi, F.: Unprecedented Ambient Sulfur Trioxide ($SO_3$) Detection: Possible Formation Mechanism and Atmospheric Implications, Environ. Sci. Technol. Lett., 7, 809-818, 10.1021/acs.estlett.0c00615, 2020.

Zhang, H., Gao, R., Li, H., Li, Y., Xu, Y., and Chai, F.: Formation mechanism of typical aromatic sulfuric anhydrides and their potential role in atmospheric nucleation process, J. Environ. Sci., 10.1016/j.jes.2022.01.015, 2022a.

Zhang, H., Wang, W., Li, H., Gao, R., and Xu, Y.: A theoretical study on the formation mechanism of carboxylic sulfuric anhydride and its potential role in new particle formation, RSC Adv., 12, 5501-5508, 2022b.

Zhang, H. J., Wang, W., Pi, S. Q., Liu, L., Li, H., Chen, Y., Zhang, Y. H., Zhang, X. H., and Li, Z. S.: Gas phase transformation from organic acid to organic sulfuric anhydride: Possibility and atmospheric fate in the initial new particle

formation, Chemosphere, 212, 504-512, 10.1016/j.chemosphere.2018.08.074, 2018.

Zhang, R., Shen, J., Xie, H. B., Chen, J., and Elm, J.: The Role of Organic Acids in New Particle Formation from Methanesulfonic Acid and Methylamine, Atmos. Chem. Phys. Discuss., 2021, 1-18, 10.5194/acp-2021-831, 2022c.

Zhong, J., Li, H., Kumar, M., Liu, J., Liu, L., Zhang, X., Zeng, X.-C., and Francisco, J. S.: Mechanistic Insight into the Reaction of Organic Acids with $SO_3$ at the Air–Water Interface, Angew. Chem., 5, 10.1002/ange.201900534, 2019.

Zhong, J., Zhu, C. Q., Li, L., Richmond, G. L., Francisco, J. S., and Zeng, X. C.: Interaction of $SO_2$ with the Surface of a Water Nanodroplet, J. Am. Chem. Soc., 139, 17168-17174, 10.1021/jacs.7b09900, 2017.

---

## Author Response (AR2)

Shi Yin

Associate Professor

College of Biophotonics,

South China Normal University,

Guangzhou 510631, P. R. China

Email: yinshi@m.scnu.edu.cn

Dear Editors and reviewers,

Many thanks for your attention to our manuscript (egusphere-2023-1649) "Organosulfate Produced from Consumption of $SO_3$ Speeds up Sulfuric Acid-Dimethylamine Atmospheric Nucleation". We appreciate the time and effort that the editors and reviewers spent on providing constructive feedback for our paper. We have carefully considered the reviewer's comments and questions and have addressed all their issues. Below, we provide response to the reviewer's comment, followed by a summary of the main revisions made in the manuscript and SI.

Best regards,

*Shi Yin*

*Referee #1*

*Reviewer's comment:*

*The authors addressed all comment and suggestions. The paper is nicely written and the conclusions are sound. I am happy to recommend the publication of the manuscript in ACP.*

Author reply:

Thanks the reviewer very much for his/her comments and efforts to improve our article.

*Referee #2*

*Reviewer's comment:*

*I apologize that my comment on the steady state concentration of $SO_3$ was not clear enough. I try to explain it better now. Jayne et al. (J. Phys. Chem. A 1997, 101, 10000-10011) report the reaction rate constant involving two water molecules. One water is the facilitator molecule that catalyzes the gas phase reaction. This is discussed in Jayne et al.,1997 and Hazra and Sinha, 2011. In the atmosphere sulfuric acid concentration is governed by the sulfuric acid formation and loss rates given by:*

*R1 $SO_2 + OH + O_2 \rightarrow SO_3 + HO_2$ k = 1.3E-12 $cm^3$ $molecule^{-1}$ $s^{-1}$*

*R2 $SO_3 + 2\,H_2O \rightarrow H_2SO_4$*

*R3 $H_2SO_4 \rightarrow$ aerosol*

*For reaction R2 the rate constant is given by Jayne et al., as $3.90 \times 10^{-41}$ exp(6830.6/T)$[H_2O]^2$, which is at T= 298: k=3.5E-31 $cm^6 s^{-1}$ . For condensation of $H_2SO_4$ on aerosol we assume a CS of 0.01 $s^{-1}$.*

*The sulfuric acid concentration at steady state can be calculated by:*

*$d[H_2SO_4]/dt = k2\ [SO_3][H_2O]^2 - CS\ [H_2SO_4] = 0$*

*This yields for sulfuric acid: $[H_2SO_4] = k2[SO_3][H_2O]^2/CS$ Eq.1*

*rearrange for $SO_3$: $[SO_3] = CS\ [H_2SO_4]/(k2\ [H_2O]^2)$ Eq.2*

*Eq.1: Assuming $[H_2O]=2E17\ cm^{-3}$ and $[SO_3] = 1E05\ cm^{-3}$ yields $[H_2SO_4] = 1.4E11\ cm^{-3}$*

*Eq. 2: Assuming $[H_2O]=2E17\ cm^{-3}$ and $[H_2SO_4] = 1E07\ cm^{-3}$ yields $[SO_3] = 7\ cm^{-3}$*

*The authors use the reaction equilibrium to demonstrate, that an assumed concentration of $[SO_3]=1E05$ $cm^{-3}$ is reasonable. They look at the reaction $H_2O + SO_3 \rightarrow H_2SO_4$ and calculate the equilibrium constant from QC data to determine a relation between $SO_3$ and $H_2SO_4$. However, the authors use the Gibbs free energy of activation barrier instead of the Gibbs free energy of formation of $H_2SO_4$. Thus, their determination of $SO_3$ or $H_2SO_4$ is wrong. Furthermore, the authors cite a few references to corroborate their assumption for $[SO_3] = 1E05\ cm^{-3}$.*

*Li et al., JACS 2018 does a calculation similar to that given above. They get $[SO_3] = 1E05\ cm^{-3}$ under the assumption of $[H_2O] = 1E15\ cm^3$. Such a water vapor concentration in the atmosphere is rather low and can only be reached at 10% RH and -20°C. In the lower atmosphere such conditions occur only at special locations and $SO_2$ concentrations might be rather low there. Thus, this value is not representative for most of the atmosphere.*

*The other papers do not give any reference, why they assume $[SO_3] = 1E05\ cm^3$. Liu et al., Zhong et al. and Tsona Tchinda et al., have all co-authors in common to Li et al. Thus, it seems that they just took over this value from each other. This does not strongly support this value.*

*The authors also mention Yao et al, PNAS 2020. As calculated above those authors should measure sulfuric acid higher than $1.0E11\ cm^{-3}$, while they measure only about $1E07\ cm^{-3}$. In Figure 3 the authors assume that there is $[SO_3] = [SA] = 1E05\ cm^{-3}$. If one uses above steady state equations for the formation of sulfuric acid this is impossible. If $[SA]$ is as low as $1E05\ cm^3$, then $[SO_3]$ is much lower and GAS would correspondingly also be lower than what the authors assumed. Thus, Figures 3 is only a theoretical exercise but not at all relevant for ambient conditions.*

Author reply:

Firstly, we would like to thank again for the reviewer's time, effort and constructive feedback spending on our paper. We agree that the reviewer's consideration and question are very important and noteworthy. We believe that the key point to the summarized issue is whether the concentrations of $SO_3$ and gas-phase organosulfate (glycolic acid sulfate, GAS) in the atmosphere considered in our work are reasonable. At present, there are indeed not many reports on direct observation data of $SO_3$ and gas-phase organosulfates in the atmosphere. Actually, we also hope to have the opportunity to draw more scientists' attention to the important atmospheric active species, $SO_3$ and gas-phase organosulfates, through this work, so that more observational data can be reported. To the best of our ability, we found a few direct observational reports of atmospheric $SO_3$ and gas-phase organosulfates, summarized as follows.

(1) Yao et al. report results from two field measurements in urban Beijing during winter and summer 2019, using a nitrate-CI-APi-LTOF (chemical ionization-atmospheric pressure interface-long-time-of-flight) mass spectrometer to detect atmospheric $SO_3$ and $H_2SO_4$ (Yao et al., 2020). They found the concentration

of $SO_3$ was in the range between $\sim 5.0 \times 10^3$ to $\sim 1.9 \times 10^6$ molecules cm$^{-3}$, and the gas-phase $H_2SO_4$ was between $\sim 4.5 \times 10^5$ to $\sim 9.0 \times 10^6$ molecules cm$^{-3}$ during the same period.

(2) Ehn et al. have made the first observation of organosulfate (glycolic acid sulfate, GAS) in the gas phase, and reported its ion concentration in the gas phase (6.7 molecules cm$^{-3}$) in the Finnish boreal forest (Ehn et al., 2010). In the same observation, the gas phase $H_2SO_4$ (SA) ion concentration was identified to be 242.7 molecules cm$^{-3}$, which is about 40 times larger than that of GAS.

(3) Le Breton et al. identified and measured 17 sulfur-containing organics (SCOs, including organosulfates, organosulfonates and nitrooxy organosulfates, GAS is one of them) at a regional site 40 km north-west of Beijing (Le Breton et al., 2018). They successfully identified a persistent gas-phase presence of SCOs in the ambient air. The mean contribution from gas-phase SCO to total SCO was found up to be 11.6 %, $\sim$ 23 ng m$^{-3}$ (approximately equivalent to $\sim 10^6$ - $10^7$ molecules cm$^{-3}$), which suggests that a significant amount of SCO is always present in the gas phase. If we assume that the GAS accounts for $\sim$ 1% of the total gas-phase 17 SCOs concentration, its gas-phase concentration can be roughly estimated to be $10^5$ molecules cm$^{-3}$.

(4) Ye et al. detected abundant oxygenated organic compounds containing two to five oxygen atoms, using an iodide chemical ionization time-of-flight mass spectrometer installed with a Filter Inlet for Gases and AEROsols (FIGAERO-I-CIMS) at Guangzhou in southern China, during the autumn of 2018 (Ye et al., 2021). They detected the ion $C_2H_3SO_6^-$ with a diurnal peak in the afternoon in both gas phase and particle phase, which ion was attributed to GAS. GAS was the main organosulfate they observed.

In our manuscript, we considered the concentration of GA in the range of $1.11 \times 10^7 - 2.72 \times 10^9$ molecules cm$^{-3}$ according to the values of series field observations (Mochizuki et al., 2019; Miyazaki et al., 2014; Stieger et al., 2021; Mochizuki et al., 2017) and the concentration of $SO_3$ in the range of $10^4 - 10^6$ molecules cm$^{-3}$ according to Yao et al.'s field observations (Yao et al., 2020). According to the estimation of thermodynamic equilibrium of chemical reaction, the concentration of GAS in the range of $2.14 \times 10^2 - 5.24 \times 10^6$ molecules cm$^{-3}$ were considered and discussed in Figure 3, Table S1, and Figure S9, respectively, in our revised manuscript.

Thanks again for the reviewer's carefully explanation of dynamic equilibrium calculation. It is clear and we agree that the calculation is right. We apologize for the mistake of using Gibbs free energy of activation barrier instead of Gibbs free energy of the formation of $H_2SO_4$. We have double checked that the right Gibbs free energy of the formation was used in the calculation of GAS equilibrium concentration in our manuscript. For reaction of $H_2O + SO_3 \rightarrow H_2SO_4$; $K_{H2O + SO3} = \frac{[H2SO4]}{[H2O][SO_3]} = e^{\frac{-\Delta G}{RT}}$; $[H_2SO_4] = K_{H2O + SO3} [H_2O][SO_3]$, the

Gibbs formation free energy of sulfuric acid is ~ 12 kcal mol$^{-1}$ (Torrent-Sucarrat et al., 2012). Assuming $SO_3$ ~ $10^5$ molecules cm$^{-3}$, $H_2O$ ~ $2 \times 10^{17}$ molecules cm$^{-3}$, sulfuric acid is estimated to ~ $5 \times 10^{11}$ molecules cm$^{-3}$. Obviously, this result is inconsistent with the observations of Yao et al. ($SO_3$, ~ $5.0 \times 10^3$ to ~ $1.9 \times 10^6$ molecules cm$^{-3}$; $H_2SO_4$, ~ $4.5 \times 10^5$ to ~ $9.0 \times 10^6$ molecules cm$^{-3}$). Our understanding is that the concentration of the reactive specie in the atmosphere may be difficult to accurately estimate from one or a few chemical reactions, due to the complexity of real environments. We think this also reminds us that when studying species whose observed concentrations are uncertain, it may be necessary to estimate a wider concentration range to discuss its atmospheric physical and chemical effects.

Overall, we understand and agree with the reviewer's questions and concerns. We agree that the exact concentration range of $SO_3$ and gas-phase organosulfate are still uncertain in the actual atmosphere, due to the lack of sufficient observational data. It is possible, under some environments and conditions, their concentrations may be low and they can be ignored. In this work, we try to theoretically discuss the neglected atmospheric chemical reactions in the atmosphere that may produce secondary products that contribute significantly to the new particle formation in the atmosphere, based on limited observational data and under certain reasonable conditions. We hope that through our theoretical work, scientists may pay more attention to the secondary products of gas phase reactions, such as organosulfates. More relevant observational studies can be reported, thus providing the possibility to explore new and more complex formation mechanisms of atmospheric new particles.

Following sentences were modified and added in our revised manuscript.

"We use the reactant concentrations of [GA] = $1.11 \times 10^7$-$2.72 \times 10^9$ molecules cm$^{-3}$ according to the values of some field observations (Mochizuki et al., 2019; Miyazaki et al., 2014; Stieger et al., 2021; Mochizuki et al., 2017). Considering field measurements (Yao et al., 2020) and theoretical investigations (Tan et al., 2022; Zhong et al., 2019; Liu et al., 2019; Tsona Tchinda et al., 2022; Li et al., 2018) of atmospheric $SO_3$, its concentration is assumed to be $10^5$ molecules cm$^{-3}$ here. Based on the above equations, the estimated concentration of the reaction product, GAS, is about $2.14 \times 10^3$-$5.24 \times 10^5$ molecules cm$^{-3}$, and GASA is about $2.30 \times 10^{-6}$-$5.62 \times 10^{-4}$ molecules cm$^{-3}$. Thus, a range of concentration for GAS, from $10^3$ to $10^5$ molecules cm$^{-3}$ as shown in Table S1, is selected for the discussion in this work." *in line 121 page 5 were modified to* "We use the reactant concentrations of [GA] = $1.11 \times 10^7$-$2.72 \times 10^9$ molecules cm$^{-3}$ according to the values of some field observations (Mochizuki et al., 2019; Miyazaki et al., 2014; Stieger et al., 2021; Mochizuki et al., 2017). Considering atmospheric $SO_3$ field measurements (Yao et al., 2020), its concentration is considered in the range of $10^4$ - $10^6$ molecules cm$^{-3}$. Based on the above equations, the estimated concentration of the reaction

product, GAS, is about $2.14 \times 10^2$-$5.24 \times 10^6$ molecules $cm^{-3}$, and GASA is about $2.30 \times 10^{-7}$-$5.62 \times 10^{-3}$ molecules $cm^{-3}$. Thus, a range of concentration for GAS, from $10^2$ to $10^6$ molecules $cm^{-3}$, is selected for the discussion in this work (Figure 3, Table S1, and Figure S9)."

"[SO$_3$] = $10^4$ and $10^6$ molecules $cm^{-3}$ are also considered and compared with the results shown in Figure 3a (as displayed in Figure S9). In the case of [SO$_3$] = $10^4$ molecules $cm^{-3}$, it is worth noting that the cluster formation rate of GAS-SA-DMA system slightly increases with the increasing [GAS] compared to that of GA-SA-DMA system with corresponding [GA], which $J_{GAS-SA-DMA}$ reaches twice the value of $J_{GA-SA-DMA}$. For [SO$_3$] = $10^6$ molecules $cm^{-3}$, the trend of this difference becomes relatively obvious, and $J_{GAS-SA-DMA}$ grows up to 2 orders of magnitude higher than $J_{GA-SA-DMA}$." *were added in line 288 page 11*.

"Le Breton et al. identified and measured 17 sulfur-containing organics (including organosulfates and GAS is one of them) at a regional site 40 km north-west of Beijing (Le Breton et al., 2018). They successfully identified a persistent gas-phase presence of organosulfates in the ambient air. The mean contribution from gas-phase sulfur-containing organics to total was found up to be 11.6 %, ~23 ng $m^{-3}$. Ye et al. also detected the ion $C_2H_3SO_6^-$ with a diurnal peak in the afternoon in both gas phase and particle phase, which ion was attributed to GAS, at Guangzhou in southern China during the autumn of 2018 (Ye et al., 2021)." *were added in line 201 page 7*.

"Note that the GAS concentration we discussed in this work are estimated from limited observational data of SO$_3$ and GA in the atmosphere. The actual atmospherics concentration of GAS still requires a large number of field observations to achieve more in-depth research." *were added in line 480 page 19*.

*The authors also claim that nucleation rates at 1.2 and 3 nm are almost similar. This is very strange. They use the Kerminen-Kulmala equation and give the used parameter values. It seems that they used the condensation sink in units of s-1. This does not conform to the units in the formula and most probably led to this strange result.*
*Alltogether, I think the authors did not convincingly show that their assumed concentration of SO3 is reasonable and supports their further conclusions. Also the nucleation rate comparison with Mount Tai is flawed because of a wrong calculation. The paper cannot be accepted as is.*

Author reply:

Thanks the reviewer for pointing out this mistake. We found the original published literatures of the revised Kerminen-Kulmala equation (Anttila et al., 2010;Lehtinen et al., 2007) and correctted the caculation as below.

$$J_x = J_1 \cdot \exp[-\gamma \cdot d_1 \cdot \frac{CS(d_1)}{GR}]$$

$$\gamma = \frac{1}{m+1}[(\frac{d_x}{d_1})^{m+1} - 1]$$

For typical atmospheric aerosols, the value of *m* can be set to -1.9. If we choose $d_1$ = 1.3 nm, $d_x$ = 3 nm, so that $J_{1.3}$ corresponds to the "nucleation rate" at 1.3 nm and $J_3$ to the "nucleation rate" at 3 nm, and set *m* = -1.9 corresponding to typical Hyytiälä event-day conditions, we have $\gamma \approx 0.5$. GR was measured to be 3.1 nm·h⁻¹ for 3.0 nm size particles during the observed events (Riipinen et al., 2007). The coagulation sink in Hyytiälä is at the level of 10⁻³ s⁻¹ (Olenius et al., 2013; Dal Maso et al., 2008). Thus we can roughly get the approximate relationship $J_3 \approx 0.5\ J_{1.3}$. Considering this different value between $J_3$ and $J_{1.3}$, Figure 7 was amended and the related description and discussion were revised in page 16 to 18 in our modified manuscript as below.

[revised manuscript text omitted]

formation rates for 3.0 nm clusters ($J_{3.0}$) relate to those for 1.3 nm clusters ($J_{1.3}$) by

$$J_x = J_1 \cdot \exp[-\gamma \cdot d_1 \cdot \frac{CS(d_1)}{GR}]$$

$$\gamma = \frac{1}{m+1}[(\frac{d_x}{d_1})^{m+1} - 1]$$

where GR is the initial cluster growth rate from 1.0 to 3.0 nm, CS represents condensation sink of clusters by preexisting particles and $\gamma$ is calculated as the function of $d$. For typical atmospheric aerosols, the value of $m$ can be set to -1.9. If we choose $d_1 = 1.3$ nm, $d_x = 3$ nm, so that $J_{1.3}$ corresponds to the "nucleation rate" at 1.3 nm and $J_3$ to the "nucleation rate" at 3 nm, and set $m = -1.9$ corresponding to typical Hyytiälä event-day conditions, we have $\gamma \approx 0.5$. GR was measured to be 3.1 nm·h$^{-1}$ for 3.0 nm size particles during the observed events (Riipinen et al., 2007). The coagulation sink in Hyytiälä is at the level of 10$^{-3}$ s$^{-1}$ (Olenius et al., 2013; Dal Maso et al., 2008). Thus we can roughly get the approximate relationship $J_3 \approx 0.5 J_{1.3}$." *were added in page S4 line 61 in our modified SI.*

Finally, please give us an opportunity to briefly explain our research purposes and ideas again. In current theoretical study we aim to reveal the potential molecular formation mechanisms of organosulfates and their potential impacts on the new particle formation, which are still much less understood. In the complex interplay of atmospheric chemistry, the role of glycolic acid sulfate in the formation of new particles has emerged as a topic of considerable importance (Long et al., 2022). Recently, Yang et al. also presented a new feasible route for the formation of organosulfates via the gas phase reactions of acetaldehyde with sulfuric acid catalyzed by dimethylamine (Yang et al., 2023). We hope more scientists will pay attention to the secondary products of gas phase reactions, such as organosulfates, through our work, and more relevant observational data can be reported. We think it will be helpful for exploring new and more complex formation mechanisms of atmospheric new particles.

We appreciate the referee's comments and questions for clarifying and improving our manuscript.